# Machine Learning-Based Computer Vision for Depth Camera-Based Physiotherapy Movement Assessment: A Systematic Review

**DOI:** 10.3390/s25051586

**Published:** 2025-03-05

**Authors:** Yafeng Zhou, Fadilla ’Atyka Nor Rashid, Marizuana Mat Daud, Mohammad Kamrul Hasan, Wangmei Chen

**Affiliations:** 1Center for Artificial Intelligence Technology (CAIT), Faculty of Information Science and Technology, Universiti Kebangsaan Malaysia, Bangi 43600, Selangor, Malaysia; yafeng.zhou.cn@gmail.com; 2Fotric Inc., 2500 Xiupu Road, Pudong, Shanghai 201315, China; 3Institute of Visual Informatics, Universiti Kebangsaan Malaysia, Bangi 43600, Selangor, Malaysia; marizuana.daud@ukm.edu.my; 4Faculty of Information Science and Technology, Universiti Kebangsaan Malaysia, Bangi 43600, Selangor, Malaysia; mkhasan@ukm.edu.my (M.K.H.);

**Keywords:** computer vision, depth camera, physiotherapy movement assessment, machine learning, systematic review

## Abstract

Machine learning-based computer vision techniques using depth cameras have shown potential in physiotherapy movement assessment. However, a comprehensive understanding of their implementation, effectiveness, and limitations remains needed. Following PRISMA guidelines, we systematically reviewed studies from 2020 to 2024 across Web of Science, Scopus, PubMed, and Astrophysics Data System to explore recent advancements. From 371 initially identified publications, 18 met the inclusion criteria for detailed analysis. The analysis revealed three primary implementation scenarios: local (50%), clinical (33.4%), and remote (22.3%). Depth cameras, particularly the Kinect series (65.4%), dominated data collection methods. Data processing approaches primarily utilized RGB-D (55.6%) and skeletal data (27.8%), with algorithms split between traditional machine learning (44.4%) and deep learning (41.7%). Key challenges included limited real-world validation, insufficient dataset diversity, and algorithm generalization issues, while machine learning-based computer vision systems demonstrated effectiveness in movement assessment tasks, further research is needed to address validation in clinical settings and improve algorithm generalization. This review provides a foundation for enhancing computer vision-based assessment tools in physiotherapy practice.

## 1. Introduction

Physical rehabilitation plays a critical role in contemporary healthcare, helping individuals regain function after injury, surgery, or chronic disease. Recent advances in machine learning-based computer vision, from convolutional neural networks (CNN) [1] to vision transformers (ViT) [2], have achieved significant success in image classification, object detection, and other visual tasks [3]. These technological advances, particularly deep learning-based computer vision, have demonstrated substantial potential in physical rehabilitation [4,5,6] and physiotherapy movement assessment [7,8,9], offering opportunities to enhance traditional physical therapy practices. For instance, in the field of telerehabilitation, Maskeliūnas et al. [10] used depth cameras combined with virtual reality technology to conduct a pilot project on post-stroke rehabilitation, generating interactive training scenarios that significantly improve patients’ balance and hand-eye coordination; Lim et al. [11] developed a robot-assisted upper limb rehabilitation system that realizes real-time monitoring and personalized feedback, effectively improving the progress of rehabilitation and reducing the problem of misstep detection; Çubukçu et al. [12] introduced depth camera technology into the home rehabilitation environment and achieved significant results in shoulder joint activity monitoring and training; and Saratean et al. [13] innovatively developed an effort-based quantitative training algorithm to optimize rehabilitation effects through real-time assessment and voice guidance, which played an important role during the COVID-19 epidemic.

The integration of depth cameras with machine learning algorithms has emerged as a promising approach for objective movement assessment in physiotherapy. These technologies enable contactless and accurate capture of patient movements, providing quantitative data for assessment and progress monitoring. Recent studies have highlighted how machine learning techniques can process this depth camera data to automate movement analysis and provide real-time feedback during rehabilitation exercises [4].

Artificial intelligence (AI) technologies, particularly deep learning (DL) methodologies, have revolutionized movement assessment in physiotherapy. As noted by Nogales, et al., “DL, the latest breakthrough in AI, can provide more effective, personalized, and efficient care, leading to improved patient outcomes” [14]. Computer vision-based modeling and analysis of human movement have garnered significant academic attention [4], offering efficient and contactless solutions for patient movement data collection.

However, implementing machine learning-based computer vision systems for depth camera-based physiotherapy movement assessment presents various challenges. These include ensuring accurate movement detection, developing robust algorithms for real-time processing, and creating systems that can adapt to different patient conditions and rehabilitation needs. Understanding these challenges and current technological solutions is crucial for advancing this field.

This systematic review examines the application of machine learning-based computer vision in depth camera-based physiotherapy movement assessment. It analyzes current methodologies, technological implementations, and challenges in using depth cameras for human movement recognition and analysis in physiotherapy settings. The review aims to provide a comprehensive understanding of existing approaches while identifying areas for future research and development.

### 1.1. Domain Characteristics

This paper focuses on machine learning-based computer vision approaches utilizing depth cameras for physiotherapy movement assessment. Depth cameras have revolutionized physical therapy by enabling accurate 3D pose capture, real-time feedback mechanisms, personalized therapy monitoring, and objective evaluation of treatment outcomes. The integration of machine learning, particularly deep learning algorithms, has proven crucial for processing depth camera data effectively. These advanced algorithms enable efficient feature extraction from sensor data, precise movement analysis, and objective evaluation, significantly enhancing treatment effectiveness. For instance, Lim et al. [15] developed an innovative Visual Feedback Balance Training (VFBT) system, which integrates depth camera and pressure pad technology with functional electrical stimulation (FES) to enhance standing balance training. This system captures joint displacement data at a sampling frequency of 20 Hz, which was validated by comparing it with data from a traditional force plate. The results showed a Pearson correlation coefficient exceeding 98%, confirming the system’s reliability and feasibility for clinical applications. Wagner et al. [16] highlighted that traditional gait analysis devices, often constrained by high costs and complexity, face challenges in broad applicability. They emphasized that introducing low-cost high-precision gait analysis using depth sensor technology offers a promising solution. With the growing demand for accessible and efficient tools in the global rehabilitation market, these findings demonstrate that innovative technologies improve diagnostic and therapeutic outcomes and significantly contribute to the accessibility and adoption of rehabilitation medicine.

We systematically analyzed the literature on depth camera-based physiotherapy movement assessment, examining the methodologies, datasets, machine learning architectures, computer vision techniques, and implementation strategies employed. This comprehensive analysis provides insights into current approaches and guidance for future developments in machine learning-based movement assessment in physiotherapy. The combination of machine learning algorithms with depth camera technology has substantially advanced the field, particularly in automated movement analysis and assessment capabilities.

### 1.2. Scope of This Review

This systematic review examined relevant articles from 2020 to 2024, focusing on the integration of machine learning-based computer vision technology in physical rehabilitation. We specifically analyzed research implementing depth cameras for physiotherapy movement assessment. The review excluded papers on virtual rehabilitation and serious games based on visual sensors, despite their use of related technologies such as human skeleton keypoint detection and target tracking, to maintain focus on clinical movement assessment applications.

Recent studies have made significant contributions to this field. Several researchers have provided crucial insights into machine learning applications in healthcare AI [14,17,18,19,20]. Notable work includes Burhani and Naqvi’s [21] research on specific applications like distal radius fracture assessment [21], while Debnath et al. [4] contributed valuable insights into computer vision applications in rehabilitation [4].

This review specifically analyzes how researchers have implemented machine learning algorithms for human motion feature extraction and recognition using depth camera data. We paid particular attention to studies that utilized depth cameras as primary data capture sensors, examining their integration with machine learning approaches for movement assessment. This focused scope allows for detailed analysis of current technical implementations and their effectiveness in physiotherapy settings.

### 1.3. Outline

As illustrated in Figure 1, this comprehensive review is structured into six main sections. Section 1 establishes the foundation of the study by introducing the domain characteristics of machine learning-based computer vision in physiotherapy, defining the scope and objectives of the review, and presenting the structural organization of this article. Section 2 details the methodological framework, beginning with the formulation of research questions, followed by comprehensive search strategies, systematic data collection procedures, and the rigorous application of the PRISMA process. Section 3 presents a systematic analysis of the results through eight research questions (RQs), examining: sensing technologies (RQ1), available datasets (RQ2), data processing methods (RQ3), algorithmic approaches (RQ4), feature extraction techniques (RQ5), application scenarios (RQ6), movement assessment targets (RQ7), and problem statements (RQ8). Section 4 provides a comprehensive discussion that synthesizes key findings, contextualizes results within existing literature, addresses open challenges, and proposes future research directions. The paper concludes with Section 5, summarizing the main contributions and implications of the study.

## 2. Methodology

This systematic review followed the Preferred Reporting Items for Systematic Reviews and Meta-Analyses (PRISMA) guidelines [22,23] to ensure transparency and reproducibility. The review protocol included the following key components: research question formulation, literature search strategy development, inclusion and exclusion criteria establishment, data extraction process, and quality assessment methodology.

The researchers selected PRISMA over other methodological frameworks such as QUOROM [24], Cochrane Handbook [25], and JBIRM [26] due to its comprehensive reporting structure and wide acceptance in both clinical and technical research domains. This choice aligned with current journal publishing standards and facilitated systematic data collection and analysis.

The research process, as illustrated in Figure 2, comprised systematic search procedures, screening protocols, and data synthesis methods. These steps ensured a rigorous and replicable review process focused on machine learning-based computer vision applications in physiotherapy movement assessment using depth cameras.

While implementing the PRISMA system evaluation, we developed a series of Python scripts for batch data processing. At the same time, we utilized Google Sheets as a collaboration platform to enable team members to work together on feature extraction, screening, evaluation, and annotation of literature. For a detailed division of labor arrangement and the specific processing flow, please refer to Section 2.4 PRISMA process.

### 2.1. Research Questions

This review explores systematic advances in recent research on computer vision for camera-depth sensor-based physiotherapy movement assignments. The primary objective is to summarize and analyze the relevant scientific literature, examine the characteristics and contributions of the work covered, and critically assess its limitations. Through this process, the paper aims to extract valuable insights and trends to guide future research. With this in mind, this review focuses on the following research questions to clearly articulate its purpose and usefulness (see Table 1).

### 2.2. Search Strategies

A literature search is essential in literature review studies to provide adequate literature support and relevant information. This paper aims to present a literature search strategy on “Applying Computer Vision for Camera Depth Sensor-Based Physiotherapy Movements Assessment”. Web of Science (WOS) https://www.webofscience.com/, (accessed on 19 January 2025), Scopus https://www.scopus.com/(accessed on 19 January 2025), PubMed https://pubmed.ncbi.nlm.nih.gov/, (accessed on 19 January 2025), and Astrophysics Data System https://ui.adsabs.harvard.edu/, (accessed on 19 January 2025), were considered in selecting the search sources, which cover a wide range of literature resources in medical and computer science-related disciplines and can provide rich research content. The articles were searched and collected from 1 January 2020 to 16 April 2024 to ensure that the latest literature findings were included. The search fields are title, keywords, and abstract.

In the literature search, the authors used different search strategies, and the search process followed a combination of computer vision and physiotherapy movements to identify and retrieve keywords. When identifying keywords, the researchers set the logical operator between the fields of computer vision and physiotherapy movements to AND to ensure that the retrieved literature covered both fields. In addition, the authors also combined the logical AND between physical therapy and movement or rehabilitation in the physical therapy domain, and the final search strategy architecture is shown in Figure 3 and in Table 2.

Combining the above strategies, a comprehensive search string was constructed by combining keywords in computer vision and physiotherapy movement. By applying this search strategy across retrieval sources, the relevant literature covering the combined application of physiotherapy movement and computer vision based on deep-camera sensors was obtained. Notably, the deletion and addition of relevant papers were carried out to obtain the final literature used for the literature review research.

### 2.3. Data Collection

A systematic literature review was conducted in databases such as WOS, Scopus, PubMed, and Astrophysics Data System using keywords related to physiotherapy, computer vision, depth camera sensors, and deep learning. Studies not focused on computer vision techniques, especially those that did not use depth camera sensors for physiotherapy movement assessment, were excluded. Critical information, such as camera types, dataset characteristics, learning architectures, and implementation strategies, was extracted.

### 2.4. PRISMA Process

This systematic review was conducted following PRISMA guidelines by a multidisciplinary team comprising three primary reviewers: a computer vision algorithm engineer (Zhou), a senior research expert in physiotherapy movement intelligence (Fadilla), and an expert in computer vision and deep learning with extensive experience in medical imaging and healthcare (Marizuana). This three-reviewer structure ensured comprehensive evaluation through majority voting and maintained methodological rigor.

The review process began with keyword definition in two domains. For computer vision, we initially identified: “Machine Learning“, “Deep Learning”, “Neural Networks“, “Computer Vision”, “Depth Camera*“, “Depth Sensor*”, “Kinect“, “RGBD”, and “RGB-D“. These terms were systematically combined with physiotherapy movement keywords using Boolean operators. Our preliminary Web of Science search yielded 211 articles. To enhance specificity, we refined our search to focus on machine learning-based computer vision applications, resulting in 97 papers.

The screening process followed a structured three-phase approach:
**Initial Screening:** Each reviewer independently assessed titles, abstracts, and keywords against our inclusion criteria focusing on machine learning-based computer vision for depth camera-based physiotherapy movement assessment. Zhou evaluated technical aspects of computer vision implementations, Fadilla assessed physiotherapy relevance, and Marizuana provided oversight on methodology and deep learning applications.**Quality Assessment:** Papers passing initial screening underwent rigorous quality assessment using standardized tools:
Risk of bias assessment using the QUADAS-2 tool for diagnostic accuracy studiesMethodological quality evaluation using the PEDro scale for clinical trialsReporting completeness assessment using the PRISMA checklist**Final Selection:** Disagreements were resolved through majority voting among the three reviewers. Each paper required approval from at least two reviewers for inclusion.

Selection criteria were established to ensure methodological rigor and relevance:**Technical Relevance:** Studies must incorporate machine learning-based computer vision and depth sensor applications in physiotherapy contexts**Methodological Quality:** Research must demonstrate robust experimental design with clear methodology, including:
Detailed description of machine learning architectures and parametersComprehensive evaluation metricsAppropriate statistical analysis**Results Validation:** Studies must include thorough validation of results with:
Clear performance metricsStatistical significance testingComparison with baseline methods**Innovation:** Research must demonstrate original contributions to the field**Reporting Quality:** Papers must meet PRISMA reporting guidelines

We implemented systematic data extraction and synthesis methods to analyze the selected studies. This included assessment of the following:Machine learning architectures and implementations;Depth camera specifications and configurations;Physiotherapy movement assessment protocols;Performance metrics and validation methods;Clinical relevance and practical applicability.

## 3. Results

A diagram explaining PRISMA can be seen in Figure 2. We found 371 papers through our search on different databases: 97 papers on WOS, 138 papers on Scopus, 130 papers on PubMed, and 6 papers on Astrophysics Data System. After removing duplicates and papers without DOIs or written in languages other than English, 238 papers were left for the first screening. A total of 23 articles were excluded at this stage, and 169 papers were excluded as they did not meet the criteria of “Applying Deep Learning-based Computer Vision for Camera Depth Sensor-Based Physiotherapy Movements Assessment”. In the second screening, 46 articles were analyzed, of which 18 papers were finally selected for this systematic review as shown in Table 3.

### 3.1. RQ 1: Sensor

Camera sensors are crucial for acquiring visual data for movement analysis and physical therapy. The type of sensor chosen directly affects the quality of the captured data, the performance of the deep learning model, and the design of the evaluation protocol. In this review, depth cameras accounted for 65.4% (see Figure 4). As shown in Figure 5, standard depth camera sensors are shown below:**Kinect series:** RGB-D camera introduced by Microsoft, which obtains accurate depth and image data through infrared ranging and color image acquisition technologies, and the main models are Kinect V1, Kinect V2, and so on.**RealSense series:** Intel’s RGB-D camera product line, using visual-inertial ranging technology, can obtain high-quality depth and motion data.**Other RGB-D cameras:** Besides the mainstream products mentioned above, third-party vendors, such as Xtion Pro, provide some RGB-D camera devices.**Ordinary RGB cameras:** These only capture color image data and must be combined with other depth estimation algorithms for data processing and analysis.

After reviewing 18 related literature (see Table 4), the Kinect series is the most widely used sensor, and 12 studies [8,11,12,13,15,16,28,29,30,31,33,34] adopted Kinect V2 as the primary Kinect camera for data acquisition. The Kinect camera has been recognized as the mainstream choice in this field because of its high accuracy and reliability. Intel RealSense series has also gained some applications; two papers [10,36] used the RealSense L515, D435i, and D415 models. RealSense cameras are technologically advanced, have excellent performance, and are expected to be used more widely in the future.

In addition, three papers [27,32,37] used ordinary RGB cameras and other RGB-D cameras (e.g., Xtion Pro, ASUS, Taipei, Taiwan) to collect data. This approach has low hardware requirements but requires the development of appropriate algorithms to process and analyze the data.

Choosing a sensor for physical therapy applications requires a combination of specific rehabilitation scenarios and technology needs, while Kinect V2 dominates current applications with its superior bone tracking capabilities and reliability for full-body motion analysis, it faces sustainability issues due to discontinuation. In contrast, Intel RealSense excels in proximity capture and fine-motion tracking and is a strong contender for home rehabilitation systems thanks to continued development support and cost advantages. Conventional RGB-D cameras, such as the Xtion Pro, offer depth perception at a lower cost, while their tracking accuracy may be slightly less than that of a dedicated system, they are still attractive in some applications. The Xbox 360 platform played an important role in early rehabilitation applications but has been replaced by more advanced technologies. Traditional RGB cameras, despite their limitations in depth perception, can still be helpful in specific scenarios, especially in resource-constrained environments, when combined with other sensors (e.g., IMUs) and appropriate pose estimation algorithms.

### 3.2. RQ 2: Dataset

In computer vision-based physiotherapy movement assessment research, datasets are a key driver for its development. High-quality and diverse raw data and labeled information are the basis for developing excellent deep-learning models. At the same time, these datasets support the ability of the models to generalize across different application scenarios, facilitating the establishment of standardized physiotherapy movement assessment methods. In the reviewed literature, researchers used a variety of data types (see Figure 6), as shown in the following:**RGB-D image/video data:** Many studies [8,10,11,16,29,32,33,35,36] have used color image and depth information captured by RGB-D cameras, usually image sequences or videos. These raw data can directly reflect the motion process and provide the basic input for subsequent motion detection and analysis.**Joint and Skeletal Data:** Several studies [12,13,27,28,34] have utilized the joint positions and skeletal information extracted by depth cameras to construct skeletal datasets. These structured data directly represents the key features of human movement and is used to model and analyze joint motion trajectories.**Combined datasets:** There are also some studies [31,32,37] that have combined RGB-D data and auxiliary data captured by other sensors (e.g., IMU) to capture the motion process from different perspectives and obtain more comprehensive and accurate motion information.

Existing studies adopted various approaches to dataset construction (see Table 5), including public datasets, self-constructed datasets, and dataset fusion. Most studies [8,10,11,12,15,16,30,32,33,34,35,37] relied on self-constructed datasets. The size of the datasets ranged from tens to hundreds of participants, with some small to medium datasets and some larger datasets, such as Girase et al. [34], which contained 411 participants. In data collection and construction, researchers generally consider data from people of different ages, genders, and health conditions. This diversity enhances the robustness of the physical therapy movement dataset and improves the ability to generalize the model.

In addition, different studies have used existing publicly available datasets to accelerate model development by utilizing existing labeled data. For instance, Raza et al. [27] used ”Multi-Class Exercise Poses for Human Skeleton“ https://www.kaggle.com/datasets/dp5995/gym-exercise-mediapipe-33-landmarks, (accessed on 19 January 2025), and Khan et al. [28] used UI-PRMD [5]. In contrast, Bijalwan et al. [29] used a fusion of multiple datasets by combining publicly available datasets (UTD-MHAD [38], mHealth [39], OU-ISIR [40], HAPT [41]) with the self-collected datasets for the construction of combinations.

During our research, we identified an important challenge in depth camera-based physical therapy exercise assessment: despite the wide variety of datasets, there is a lack of standardization and interoperability. Due to ethical and privacy considerations, most current computer vision-assisted physical therapy research relies primarily on self-constructed datasets. Researchers typically construct or integrate multiple datasets based on specific needs, which cover different groups and sizes of participants and contain multimodal data from multiple sensors, while these datasets provide some foundation for deep learning model training and physical therapy exercise analysis, there is still a lack of standardized datasets to support direct comparisons and meta-analyses of research results. These issues can be addressed in the future by establishing standardized guidelines and developing advanced annotation tools to enhance the utility of the datasets.

### 3.3. RQ 3: Data Processing

Data processing plays a key role in depth camera-based physiotherapy movement analysis studies and directly affects the quality and precision of subsequent analyses. As Table 6 shows, through a systematic review of 18 works in the literature, the author can summarize several major types of data processing methods and techniques. First, skeletal data extraction and processing are the basis of most studies, e.g., [12,34,36] used Kinect SDK 2.0 and the OpenPose library to extract human skeletal data from raw depth images, respectively, which provides structured input for subsequent analysis. Second, to ensure the consistency and comparability of the data, some studies such as [16,28] performed coordinate system transformation and alignment operations, which helped to eliminate errors caused by different devices or shooting angles.

Data standardization and normalization is another common processing step, such as the min–max normalization method used in [29], which helps to eliminate the effects of different scales and allows various features to be compared on the same magnitude. To improve data quality, some studies have used filtering and noise removal techniques, such as the Kalman filter and low-pass Butterworth filter used in [30], which effectively remove noise and improve signal quality. Feature extraction and selection also play an important role in machine learning, as evidenced by several articles in the literature [27,33]. These techniques help to reduce data dimensionality and improve the efficiency and generalization of the model.

For studies involving multimodal data, data synchronization and fusion become key issues. In [31,32], the authors explored the fusion of accelerometer and gyroscope measurements and visual synchronization via timestamps, respectively. In addition, to increase the diversity of training samples and improve the robustness of the model, ref. [35] employed data augmentation techniques, which are particularly useful in deep learning model training to alleviate the problem of insufficient data effectively. Some studies such as [28] have also mentioned the processing of feature transformations, including operations such as dimensionality reduction and feature combination, which help to extract more meaningful feature representations.

These diverse data processing methods cover the whole process from raw data acquisition to feature extraction, which improves the data quality and provides more reliable and effective inputs for subsequent algorithmic models. However, from the reviewed literature, most studies focus more on implementing specific tasks without fully exploring the impact of data processing methods on the results and the potential optimization space. This suggests that the standardization of data processing techniques has not yet been fully achieved, and future in-depth exploration in this area is needed.

Researchers can further combine advanced techniques, such as automated feature engineering with multimodal data fusion methods, to reduce data noise and enable real-time data processing in a broader range of clinical scenarios. In addition, several studies in the literature (e.g., [30,36]) have shown that data fidelity and maximized extraction of helpful information are particularly critical in practical applications. Therefore, developing more generalized and adaptable processing flows will be key to improving the accuracy and applicability of depth camera-based motion analysis in the future.

### 3.4. RQ 4: Algorithm

As Table 6 shown, each of these algorithms excels in different application scenarios, collectively contributing to the field’s rapid development. In the research field of deep camera-based physiotherapy movement assessment, the selection and design of algorithms are crucial and directly affect the accuracy and efficiency of movement recognition, assessment, and analysis. Based on a systematic review of existing literature, as shown in Figure 7, algorithms can be broadly categorized into three groups: traditional machine learning algorithms, deep learning algorithms, and dedicated algorithms for specific tasks:Traditional machine-learning algorithms have been widely used in several studies. For instance, random forest (RF) [42], logistic regression (LR) [43], support vector machine (SVM) [44], and principal component analysis (PCA) [45] are favored for their interpretability and computational efficiency. Ref. [27] utilized RF, LR, and LSTM algorithms for human posture estimation. At the same time, PCA, nonlinear PCA (NLPCA), and LR are combined to identify movement strategies in patients with low back pain [30]. Raza et al. [27] used RF, LR, and LSTM algorithms for human pose estimation in structured lower limb motion datasets where the RF approach achieved state-of-the-art with a high-performance score of 99.8%. Ref. [34] conducted a comparative study on the performance of different machine learning algorithms for behavioral classification tasks. Using the combined feature set of pose, pose derivative, and dynamic features, the MLP achieved the best classification accuracy of 52.3%, followed by RF at 51.8% and SVM at 47.2%. The study demonstrated that as the feature set gradually expanded from single pose features to complete pose-derivative dynamic feature combinations, all algorithms showed significant performance improvements, highlighting how feature combination diversity plays a crucial role in enhancing classification results.With the rapid development of deep learning techniques, more studies are using deep neural network models. Architectures such as convolutional neural networks (CNNs), recurrent neural networks (RNNs), and long short-term memory networks (LSTMs) show significant advantages when dealing with complex time-series motion data. For example, Maskeliūnas et al. [10] showed that CNNs achieved recognition accuracies ranging from 60.7 to 93.8% in human posture and motion analysis tasks, while traditional Random Forest (RF) methods achieved accuracies of 86–100%, while a hybrid deep learning (HDL) model combining CNN, RNN, and CNN-GRU effectively improved the detection and recognition accuracy of upper limb rehabilitation movements, the CNN model alone reached 98%, and the accuracy of the hybrid models CNN-LSTM and CNN-GRU reached 99% and 100%, respectively [29]. In another study, researchers developed an innovative approach combining convolutional neural networks (CNN) and random forest (RF) classifiers to estimate the human body’s center of mass (CoM). This hybrid method demonstrated remarkable performance, achieving high sensitivity and specificity rates exceeding 80% for four-level classification and over 90% for binary classification [35]. These deep learning methods automatically extract deep features from raw data, significantly reducing the workload of manual feature engineering and improving model generalization, which is particularly suitable for processing large-scale and high-dimensional visual data.In addition, some researchers have developed specialized algorithms for specific physiological therapy tasks. For example, imitation learning is used to achieve adaptive learning for multifunctional upper limb rehabilitation [11], while the effort-based parameterization method (EBPM) provides a theoretical basis for a home rehabilitation guidance system [13]. Though narrow in application, these specific task-oriented algorithms often provide precise and efficient solutions, reflecting the researchers’ deep understanding and innovative thinking about actual clinical needs. It should be noted that some studies have adopted the algorithm fusion strategy to utilize the advantages of different algorithms fully. For example, combining SVM, RF, multilayer perceptrons (MLPs), and cascaded convolutional neural networks (CCNNs) with semi-supervised learning and a traceless Kalman filter identifies the key factors of pathological movements. Girase et al. [34] evaluated the performance of different algorithms on a classification task through a five-fold cross-validation experiment based on data collected in clinical practice. For non-temporal data, random forest achieves an optimal accuracy of 51.8% when fusing features (P+PD+D). For temporal data, US CNN + MLP performs best with an accuracy of 73.4% using the complete feature set, outperforming DTW (63.0%) and ResNet (71.6%). With the enrichment of feature combinations (P to P+PD to P+PD+D), each algorithm’s performance generally improves, highlighting the positive effect of feature diversity on the classification effect. Wagner et al. [16] integrated various algorithms, including KD, CH, and FV, with the Zebris FDM platform to estimate gait parameters accurately. Among these methods, the KD algorithm demonstrated superior performance with the highest accuracy rate of 81.8%. This fusion strategy improves the performance and stability of the model, providing new ideas for solving complex physiological treatment problems.

Figure 8 visualizes the distribution of the proportion of research in the current literature among the three types of algorithms: traditional machine learning algorithms account for 44.4%, deep learning algorithms account for 41.7%, and specialized task algorithms account for 13.9%. This study indicates that traditional and deep learning algorithms are the mainstay of current research, while specialized task algorithms focus more on specific clinical needs. As described in Table 7:**Traditional Machine Learning Algorithms:** These algorithms perform well on small, structured datasets such as Sit-to-Stand or TUG test tasks. However, they have limited performance on high-dimensional and nonlinear data. Their main advantage is their high interpretability, making them valuable in clinical scenarios requiring a clear basis for decision-making.**Deep Learning Algorithms:** With their ability to process large-scale and complex data, deep learning algorithms are excellent in analyzing time-series motion data, such as high-precision upper limb rehabilitation task detection using CNN and RNN. However, they require high computational resources and lack interpretability. In the future, clinical trust can be improved by introducing interpretable AI methods such as SHAP or LIME.**Dedicated Algorithms:** Algorithms designed for specific tasks show high accuracy and relevance, e.g., the application of imitation learning in adaptive rehabilitation. However, due to their high specificity, they may need redesigned when scaling to a wide range of scenarios.

In general, movement analysis based on depth research cameras for physiological treatments has shown a trend of diversification, specialization, and convergence in algorithm selection and design. Researchers flexibly utilize existing machine learning and deep learning algorithms and develop innovative solutions based on specific application scenarios. This diversified algorithmic application strategy has effectively promoted technological advancements and improvements in clinical practice. At the same time, focusing on the development of the algorithms themselves, data pre-processing and post-processing play equally important roles in the application of algorithms. Precisely, coordinate system transformation [16], feature selection and hyperparameter tuning [27], min–max normalization [29], Kalman filtering and Butterworth low-pass filtering [30] improve the quality of the data and performance of the algorithms. The OpenPose library, for instance, is used to process video frames from RGB sensors for joint estimation [36]. These processing techniques complement the core algorithm, forming a complete analysis flow.

In the research of depth camera-based physiotherapy movement assessment, in addition to the optimization and innovation of the algorithms themselves, the following research directions are worth exploring in depth:**Model Interpretability and Trustworthiness:** Although deep learning algorithms perform well in complex data scenarios due to their powerful feature extraction capabilities, their inherent “black box” characteristics lead to insufficient model interpretations, which triggers a crisis of trust in clinical application scenarios. To address this challenge, future research needs to find a balance between model performance and interpretability. Specifically, explainable AI (XAI) techniques will be introduced to elucidate the decision rationale through feature visualization tools or auxiliary explanatory models. For example, technologies such as SHAP (Shapley Additive Explanations) and LIME (Local Interpretable Model-agnostic Explanations) can effectively enhance the interpretability of deep learning models, making them more likely to gain the trust of clinical workers.**Generalization Capability and Task Customization:** Although the current dedicated algorithms perform well in specific scenarios, their generalization ability remains to be verified. Future research should focus on developing generalized algorithms that can adapt to diverse tasks while maintaining high performance. Through techniques such as transfer learning and multi-task learning, the cross-scenario adaptability of algorithms can be enhanced while maintaining the advantages of task-customized design to solve real-world problems effectively in specific scenarios.**Data Scarcity and Migration Learning:** The acquisition of physical therapy data faces the dual challenges of high labeling costs and strict privacy protection requirements. In this context, migration learning and few-shot learning provide feasible solutions to the data scarcity problem. For example, migrating deep learning models pre-trained on large-scale datasets to new rehabilitation tasks can significantly reduce the demand for the amount of data in the target domain while guaranteeing the performance of the models.**Real-Time Processing and Edge Computing:** With the rapid development of the Internet of Things (IoT) and edge computing technologies, developing efficient and lightweight real-time processing algorithms has become the key to meeting clinical needs. Realizing low-latency and high-precision real-time feedback, especially in home rehabilitation scenarios, puts higher requirements on algorithms. Future research should focus on optimizing the efficiency of the algorithms so that they can run stably on edge devices and provide immediate motion assessment and feedback to patients.**Ethics and Privacy Protection:** Patient privacy protection and data security are important topics that cannot be ignored when applying physical therapy algorithms. Future research must introduce privacy-preserving techniques, such as federated learning during data collection and processing, to improve model performance while ensuring data security. In addition, improving the transparency and fairness of algorithms is also an important direction in enhancing their ethical acceptability.

### 3.5. RQ 5: Feature

As shown in Table 8, the summary of current research and innovations in physiotherapy and rehabilitation highlights the integration of advanced technologies and methods. Incorporating computer vision technology, depth cameras, and other sensors has been pivotal in enhancing the effectiveness and accuracy of rehabilitation treatments. These technologies offer cost-effective training feedback, improve gait assessment, increase the accuracy of human posture estimation, enhance patient engagement and the precision of posture and movement analysis, enable personalized adaptive learning, accelerate post-stroke movement assessment, and support clinical decision-making. Additionally, these technologies are used to develop home rehabilitation protocols and remote physiotherapy guidance systems, promoting continuous patient care and recovery. This systematic review categorizes and discusses the features of using depth camera sensors in physiotherapy movement assessment, focusing on the following aspects:**Integration of Depth Sensors with Other Technologies:** Research on integrating depth cameras with other sensors, such as pressure mats, is crucial in physiotherapy movement assessment. For example, ref. [15] demonstrated efficient balance training feedback by combining depth cameras and pressure mats.**Gait and Posture Assessment:** Gait and posture assessment are critical in physiotherapy. Several studies explore the application of depth sensors in these areas. For instance, ref. [16] utilized depth sensors to improve gait assessment accuracy, enhancing diagnosis and treatment. Ref. [33] applied Kinect SDK skeletonization to assess the straight leg raise accurately, aiding in lumbar condition diagnosis. Ref. [34] used machine learning to automatically detect and classify pathological movements from sit-to-stand transitions.**Upper and Lower Limb Rehabilitation:** Upper and lower limb rehabilitation is a key research direction in physiotherapy. Ref. [11] explored a personalized adaptive learning system for upper limb rehabilitation, improving patient outcomes. Ref. [36] developed a method using RGB-D sensors for precise joint angle estimation in-home rehabilitation. Ref. [13] implemented a Kinect-based remote physiotherapy guidance system, promoting continuous care.**Applications of Machine Learning and Deep Learning:** Machine learning and deep learning are widely applied in physiotherapy assessments. Ref. [27] improved human pose estimation using the LogRF and random forest algorithms. Ref. [28] introduced a hybrid quantum neural network to enhance the speed and accuracy of post-stroke movement assessments. Ref. [30] utilized unsupervised learning to analyze motion capture data, identifying movement strategies in low back pain patients. Ref. [29] combined deep learning models to enhance spatiotemporal feature modeling in stroke rehabilitation.**Home Rehabilitation and Remote Monitoring:** Home rehabilitation and remote monitoring are current research hotspots. Ref. [31] proposed a home rehabilitation protocol for post-knee replacement using convenient technology. Ref. [12] developed a system for dynamic monitoring and correction of shoulder movements using Kinect. Ref. [8] used RGB-D cameras to analyze compensatory trunk movements, improving upper limb rehabilitation strategies.**Clinical Applications and Validation:** Several studies validate the effectiveness of depth sensors in clinical settings. Ref. [32] accurately assessed movement limitations caused by spinal arthritis using RGB-D cameras, supporting better clinical decision-making. Ref. [37] demonstrated the reliability and speed of kinematic assessments using RGB-D cameras in clinical environments.

In summary, computer vision-based physiotherapy movement assessment using depth camera sensors have shown diverse applications and significant advancements. These technologies not only demonstrate great potential in enhancing diagnostic accuracy, personalized rehabilitation, and patient engagement but also pave the way for more effective and accessible physiotherapy solutions. Studies indicate that depth sensors are widely applied in gait and posture assessment and upper and lower limb rehabilitation, and, when combined with machine learning and deep learning technologies, have achieved breakthroughs in home rehabilitation and remote monitoring. These studies cover balance training, virtual reality integration, and home rehabilitation, providing real-time, accurate, and personalized feedback mechanisms that improve treatment outcomes and patient participation. Future research should focus on integrating these features into comprehensive systems to enhance further diagnostic accuracy, movement assessment speed, and home rehabilitation efficacy, promoting more efficient and convenient rehabilitation practices. In general, applying depth cameras and advanced algorithms brings innovative solutions to physical therapy, significantly improving the efficiency and coverage of rehabilitation training.

### 3.6. RQ 6: Scenario

The movement analysis based on depth cameras for physical therapy shows significant value and potential in three major scenarios (see Table 9): remote, clinical, and local (see Figure 9). In remote scenarios, the technology breaks through geographical limitations and enables patients to receive real-time rehabilitation guidance and assessment at home, improving the accessibility and continuity of rehabilitation services; in clinical scenarios, it provides medical professionals with accurate exercise data and objective assessment tools, which help to formulate personalized and efficient treatment plans; and in local scenarios, such as at home or in community-based rehabilitation centers, the technology supports autonomous training and daily monitoring and enhances patients’ self-management ability. The integration of these three scenarios optimizes the allocation of rehabilitation resources and realizes an all-round multilevel rehabilitation care system, significantly improving the overall effect of physical therapy and patient experience. With the advancement of technology and in-depth clinical practice, this multi-scenario application mode is reshaping the traditional rehabilitation concept and promoting the development of physical therapy in the direction of intelligence, personalization, and popularization.

In a remote scenario, the main objective is to provide patients with a convenient home rehabilitation program. With the development of telemedicine technology, remote physical therapy movement assessment based on depth cameras has become a reality. For example, ref. [10] described BiomacVR, a virtual reality (VR)-based rehabilitation system that combines a VR physical training monitoring environment and upper limb rehabilitation technology for precise interaction and improves patient engagement, which is applied to a real-time physical therapy sports wellness system for telerehabilitation. In [11], the authors proposed an adaptive learning system based on imitation learning for multi-purpose upper extremity rehabilitation that allows patients to perform rehabilitation at home. In [12], the authors examined the development of a Kinect 2 sensor-based telerehabilitation system that observes and evaluates exercise in patients with shoulder impairments through a web application used for communication between the patient and the therapist and a console application that helps the patient perform the exercise correctly. Ref. [13] proposed an approach based on effort parameterization for monitoring a home rehabilitation system to ensure correctness and adherence to rehabilitation exercises.

In clinical scenarios, physiotherapy movement assessment research focuses on accurately analyzing and assessing patients’ motion status to provide key data support for clinical diagnosis and treatment. For example, ref. [16] realized the accurate analysis of patients’ gait through the estimation of gait parameters, which effectively assists clinical diagnosis and the formulation of treatment plans. Due to the wide application of artificial intelligence technology in this field, for example, ref. [10] utilized neural network algorithms to observe human skeletal motion through visible information, which can accurately analyze patient posture and movement patterns. In addition, ref. [29] applied deep learning techniques for the detection and recognition of detecting and recognizing upper limb rehabilitation exercises, which helps clinicians assess the progress of the rehabilitation of patients. In disease-specific studies, such as [30], machine learning methods have been applied to identify exercise strategies for patients with low back pain, providing a scientific basis for developing clinical treatment programs. Ref. [32], which validated and analyzed patients’ trunk movement limitations by synchronizing and visualizing datasets, helps clinicians gain insights into patients’ movement abilities and limitations. In [33], the authors employed advanced detection and tracking techniques, combining calibration, skeletonization process, and feature extraction, to achieve monitoring and analysis of key movements in the rehabilitation process, providing detailed movement data support for clinical decision-making. Ref. [34] applied semi-supervised learning algorithms to estimate the joint center position through the standard Kinect 2 body tracking library, successfully identifying and classifying critical factors of pathological movements and providing more accurate data support for rehabilitation treatment.

In local scenarios, physiotherapy exercise and assessment research focus on using advanced algorithms and data processing techniques to automate the evaluation of patient rehabilitation training and posture recognition and improve rehabilitation effects and patient compliance. For example, ref. [15] provided visual feedback to patients through the acquisition and processing of joint displacement data in real-time to help them perform effective balance training at home. Ref. [27] applied AI algorithms combined with MediaPipe pose labeling, feature selection, and hyperparameter tuning to achieve a high-precision estimation of human posture, which provides important data support for rehabilitation training. Ref. [28] realized automated evaluation of exercises through high-quality neural network alignment of length and center and feature transformation, which significantly improves the efficiency and effectiveness of rehabilitation training. In rehabilitation after specific surgeries, ref. [31] incorporated accelerometer and gyroscope measurement techniques to perform home rehabilitation training evaluation after total knee replacement. This makes it possible to monitor patient rehabilitation progress in the home environment. Ref. [35] accurately estimated the patient’s center of mass position through data augmentation techniques, providing a scientific basis for balance training and assessment. Ref. [36] achieved an accurate estimation of joint position by processing video frame data from RGB sensors, providing strong support for motion analysis. In addition, ref. [37] created virtual skeletal representations to assess patients’ ability to perform functional tasks, further extending the scope and depth of local rehabilitation assessment. These studies fully demonstrate that patient rehabilitation training can be effectively monitored and evaluated in local scenarios with advanced algorithms and data processing techniques, improving the rehabilitation effect and significantly enhancing patient compliance.

### 3.7. RQ 7: Target

In physiotherapy movement assessment, the selection of an appropriate target for the study is critical. This selection is directly related to the type of visual data to be captured and its depth, which affects the design and application of deep learning models. By carefully selecting targets for the human body, researchers can ensure that the acquired movement data is pertinent and complete, providing high-quality input for movement recognition and assessment.

As depicted in Figure 10, the existing literature shows that researchers generally focus on the main body parts, such as the entire body, the upper limb, and the lower limb, as well as specific joint parts, such as the knee, ankle, and shoulder (see Table 4). For instance, five studies [13,15,27,28,35] have delved into full-body movement recognition and evaluation and focused on how to capture body movement data using a depth camera and analyze the data using deep learning models.

The upper limbs are another key focus area, with four studies [8,10,11,29] concentrating on arm and elbow joint movements to support rehabilitation efforts. These studies focused on capturing movement data at the arm and elbow joints to guide upper limb rehabilitation. Similarly, the lower limbs have received extensive attention, including the foot, ankle, knee, and hip. Six publications [16,31,33,34,36,37] have also been aimed at providing assessment and guidance for lower limb rehabilitation.

Additionally, three studies [30,32,34] emphasize the lower back and trunk, which are vital for assessing overall physical mobility. These investigations underline the significance of trunk and lumbar regions in evaluating biomechanical function and rehabilitation outcomes.

Beyond isolated target regions, exploring the biomechanical synergies and kinematic couplings between different body parts offers a more integrated perspective on movement dynamics. For instance, upper and lower limb coordination during complex physiotherapy exercises can reveal compensatory patterns or inefficiencies. Similarly, trunk and lower limb interactions during gait or balance tasks are critical for holistic movement quality assessment. This interconnected analysis can inform the development of comprehensive movement assessment models, enabling the design of physiotherapy protocols that consider the cascading effects of rehabilitation on multiple body parts. Such models could enhance our understanding of how targeted rehabilitation impacts overall biomechanics, improving intervention effectiveness.

In summary, the reviewed literature addresses movement recognition and assessment for the whole body, key limbs, and specific joints, emphasizing the importance of targeted selection in efficient data acquisition and analysis. By integrating an understanding of biomechanical synergies and kinematic couplings, researchers can develop more holistic and practical deep-learning models, fully leveraging the potential of depth cameras and other sensing technologies for physiotherapy applications.

### 3.8. RQ 8: Problem Statement

As shown in Table 10, most studies indicate that using computer vision and depth sensor technology for patient movement analysis and rehabilitation training has become a significant development direction in modern physiotherapy and assessment. However, the problem statements from a systematic review reveal numerous challenges in applying current technologies and methods. To better understand the bottlenecks in current research and future development directions, this paper categorizes and discusses these problems as follows:
**Equipment and Feedback Mechanism Issues:** Several studies [13,15,35] highlight that current balance training and motion analysis equipment is often expensive and bulky, limiting its use in resource-constrained clinical environments. There needs to be more effective on-demand balance assessment tools in physiotherapy, further restricting treatment flexibility and real-time feedback capabilities.**Data Utilization and Diagnostic Accuracy Issues:** Traditional gait analysis and kinematic assessment methods need to effectively utilize depth sensor data, leading to decreased diagnostic accuracy [16,37]. For example, conventional methods cannot accurately capture the straight leg raise motion [33], complicating lumbar assessments. Furthermore, home rehabilitation methods inaccurately estimate joint angle ranges [36], which affects patient treatment outcomes.**Accuracy Issues in Posture Estimation and Motion Analysis:** Posture estimation in physiotherapy often lacks accuracy [10,27], impacting the correction of exercises and rehabilitation effectiveness. Traditional motion analysis tools lack precision and interactivity, failing to meet the demands of efficient rehabilitation.**Personalization and Dynamic Adaptability Issues:** Standard upper limb rehabilitation devices and post-stroke assessment systems lack dynamic adaptation to patient progress [11,28], limiting their effectiveness. Existing shoulder rehabilitation systems lack precise and interactive exercise monitoring [12], making personalized treatment difficult. Inadequate motion data analysis limits low back pain treatment strategies [30].**Spatiotemporal Feature Modeling Issues:** Insufficient spatiotemporal feature modeling in stroke rehabilitation [29,31] affects the effectiveness of rehabilitation exercises. Moreover, current methods also fall short in analyzing compensatory trunk movements [8], further impacting upper limb rehabilitation outcomes.**Pathological Diagnosis and Assessment Tool Issues:** Current automated diagnostic tools for spine, hip, and knee pathologies [34], as well as tools to assess movement limitations in ankylosing spondylitis [32], are inadequate. These issues indicate that the existing automated diagnostic and assessment tools still need to meet clinical needs and require further development and optimization.

Future research and technological innovations in computer vision-based physical therapy exercise assessment can focus on several key directions to overcome the challenges of ethics, privacy, model interpretability, clinical trust, and generalization capabilities. First, in terms of privacy protection, differential privacy, federated learning, and encrypted computing techniques can be employed to ensure data security. At the same time, developing transparent, ethical guidelines and compliance frameworks can enhance patient trust in the technology. Second, in terms of model interpretability and clinical trust, developing visual AI decision tools and integrating multimodal data such as vision, EMG signals, and pressure sensor data improves the reliability and interpretability of results. In addition, in terms of generalization capability and applicability, migratory learning, data augmentation, and personalized model fine-tuning techniques are utilized to adapt the system to different patient populations and clinical environments.

Meanwhile, emerging sensor technologies such as high-resolution 3D cameras, flexible sensors, and smart wearable devices are being explored to provide more accurate support for motion capture. Regarding algorithm design, spatiotemporal perception models, few-sample learning, and self-supervised learning are used to solve the problem of limited data volume, and the real-time performance of the model is optimized by lightweight and edge computing. Finally, user-friendly interactive interfaces and personalized feedback systems are developed, which are combined with virtual reality or augmented reality technologies to provide collaborative and interactive physical therapy experiences for patients and clinicians. These innovative directions will advance the field and significantly improve the technology’s effectiveness and trust in practical applications.

## 4. Discussion

### 4.1. Summary of Key Findings

This systematic literature review provides an in-depth exploration of the application of depth camera-based computer vision techniques in physiotherapy movement assessment, revealing the following key findings:**Diversification of application scenarios:** The study shows that depth camera technology presents a diversified application trend in physical therapy, which is mainly distributed in three major scenarios: remote (16.6%), clinical (27.8%), and about local (50%). This diversified application mode optimizes the allocation of rehabilitation resources and significantly promotes the development of physical therapy in the direction of intelligence, personalization, and popularization. It is particularly noteworthy that in remote and home rehabilitation, depth camera technology effectively breaks through geographical limitations, dramatically improves the accessibility and continuity of rehabilitation services, and provides patients with more flexible and convenient treatment options.**Dominance of sensor technology:** Regarding sensor selection, depth cameras dominate (65.4%) in physical therapy applications. In particular, the Kinect and RealSense series have been widely adopted for their outstanding reliability and technological maturity. This trend fully reflects the unique advantages of depth sensors in providing markerless, non-contact human movement information, which provides a solid technological foundation to drive the development of personalized rehabilitation.**Diversity of data types and processing techniques:** Regarding data types and processing, the study shows that RGB-D and skeletal data occupy 55.6% and 27.8%, respectively, constituting the main data types. The data processing techniques are diversified, ranging from skeletal data extraction and coordinate system transformation to feature selection and noise reduction. RGB-D data have unique advantages in motion analysis and rehabilitation research, especially in enhancing data dimensionality and real-time performance. However, their limitations (e.g., occlusion problems and light dependence) may affect data reliability and application effects in specific scenarios. Skeletal data excel in motion analysis and rehabilitation due to their structured and efficient nature and are particularly suitable for real-time application scenarios. However, they also face limitations like occlusion issues and a lack of semantic information about the environment. To address these challenges, researchers can make up for the shortcomings by combining skeletal data with multimodal data (e.g., RGB-D data) and advanced algorithms, such as deep learning, while focusing on developing more robust algorithms to fully utilize the potential of the data for more comprehensive and accurate motion analysis.**Balance and trend of algorithm selection:** Regarding algorithm selection, traditional machine learning algorithms (44.4%) and deep learning algorithms (41.7%) show a relatively balanced proportion of usage. This balance reflects the researchers’ efforts to seek an optimal balance between the pursuit of model interpretability and performance. Meanwhile, deep learning algorithms show increasingly significant advantages in processing complex time series motion data, further indicating that future research will increasingly adopt it.**Distribution of research focus:** The research primarily focuses on the rehabilitation assessment and motion analysis of the upper limbs, lower back, knees, and the entire body. These body regions are critical for daily activities and quality of life, making them central academic and clinical interest areas. Studies on upper limbs often target functional recovery in post-stroke patients (e.g., [8,28]). At the same time, lower back research emphasizes functional testing and efficacy evaluation for nonspecific lower back pain (e.g., [32,37]). Knee rehabilitation research is concentrated on postoperative recovery monitoring (e.g., [31,36]), and full-body assessments leverage multimodal sensors for comprehensive analysis (e.g., [11,35]). This distribution highlights the practical demands of rehabilitation research and validates the utility and advantages of depth camera technology in capturing diverse human movements. Furthermore, these studies illuminate potential directions for future research, suggesting an expansion of depth camera applications to underexplored body regions and more complex functional scenarios, enabling a more holistic approach to movement rehabilitation assessment.

In summary, the findings of this literature review not only systematically elucidate the current state of the application of depth camera technology in physical therapy but also provide a clear direction for future research. These findings highlight the critical role of technological innovation in driving personalization, precision, and accessibility in physical therapy and the need for interdisciplinary collaboration in advancing the field.

### 4.2. Comparison with Existing Literature

This systematic literature review resonates with the existing literature and provides essential extensions and additions through a multidimensional analysis. Regarding the range of technical applications, this study extends the work of [4] emphasizing computer vision’s potential in evaluating human movement. In contrast, this study quantifies the trend of depth camera applications through systematic analysis and provides a specific distribution of application scenarios and evaluation of their effects. This in-depth analysis provides a more precise direction guide for future research.

This study presents a more comprehensive and dynamic picture of technology evolution than [46]. Although Kinect still dominates, our study reveals that emerging depth camera technologies (e.g., Intel RealSense) are gaining more and more attention. This finding reflects the trend of technology diversification and provides researchers and clinical practitioners with a broader perspective on technology selection.

Regarding AI applications, this study complements the work of [20,47], while these studies focused on using AI in skeletal data analysis and holistic physical rehabilitation, the present study provides a more comprehensive view, covering the entire process from data collection to algorithm selection. Our findings highlight the applicability of AI techniques in different assessment tasks, providing essential insights into the optimization and selection of AI systems.

In terms of data processing, this study extends the work of [48]. We focus on common RGB-D and skeletal data and also delve into various data processing techniques, such as coordinate system transformations and feature selection. This comprehensive analysis provides researchers with a more decadent choice of data processing strategies, which helps improve the accuracy and efficiency of data analysis.

In terms of the assessment system framework, compared to the AI-assisted physical therapy assessment framework proposed by [49], this study provides a more systematic and comprehensive assessment system. We focused on emotion detection and movement recognition. We explored how to simulate the clinical assessment process, laying a theoretical foundation for building a better AI-assisted rehabilitation system.

In addition, this study complements the research of [50] on applying depth sensors in in-home activity monitoring for older people. Our study covers home, clinical, and remote rehabilitation scenarios, providing a more comprehensive analysis of application scenarios and helping to promote the flexible application of depth camera technology in different rehabilitation settings.

Overall, this study significantly contributes to specific application scenarios and effectiveness assessment of depth camera technology, diversity and applicability analysis of data processing methods, application and selection considerations of AI algorithms in different assessment tasks, and proposes a more systematic and comprehensive review framework for assisted rehabilitation assessment. These findings fill the gaps in the existing literature and provide a solid theoretical foundation for interdisciplinary collaboration and technological innovation in physical therapy. The systematic analysis of this study provides a clear direction for future research, especially in improving the applicability of the technology, data processing efficiency, and algorithmic accuracy, which is expected to promote the development of physical therapy technology in the direction of being more innovative, more accurate, and more personalized.

### 4.3. Open Issues and Challenges

As shown in Table 11, there are several limitations in the research on using computer vision with depth sensors for physiotherapy movement assessment. Most studies have been conducted on healthy participants or small datasets, lacking validation on patients with various pathological conditions. Algorithms and systems are often tested in laboratory settings, which do not reflect real-world complexity. The high computational resources required for deep learning models and virtual reality systems, along with patient discomfort, limit their widespread application. Sensor resolution and reliability issues, inaccuracies in Kinect SDK capture, and sensitivity to environmental interference and specific clothing also impact assessment results. Many studies do not consider the sustainability of long-term home rehabilitation plans and the complexity of dynamic movements. For instances:**Data Samples:** Several studies are limited to testing on healthy participants or small demographic groups, failing to include patients with spinal cord injuries or varying cognitive impairments, which restricts the generalizability of the findings [15,30].**Experimental Environment:** Many studies are conducted in controlled laboratory settings, which do not adequately reflect real-world diversity and complexity. Some algorithms perform well in controlled environments but may significantly underperform in cluttered or complex real-world settings [16,27,35].**Technical Solutions:** Depth sensors like Kinect have limited resolution and field of view, potentially failing to capture fine joint movements. The accuracy of these sensors can be affected by wearable interference and environmental factors [13,32,33,36,37].**Algorithm Application:** Many current algorithms are validated only on static exercises, not sufficiently exploring the complexity of dynamic movements. Although deep learning and machine learning models show some effectiveness, they require significant computational resources and are limited by the size of datasets, which can affect model generalization [12,29].**User Experience:** Virtual reality systems may cause discomfort or dizziness in some patients, limiting their broader application. Studies often fail to consider the feasibility of long-term adherence to home rehabilitation programs, impacting practical effectiveness [10,31].

In summary, while depth cameras and computer vision technologies offer new possibilities for physiotherapy, the aforementioned limitations need to be addressed in future research. This requires technological innovation, interdisciplinary collaboration, and careful consideration of cost-effectiveness and user experience. In addition, these limitations highlight the need for future research to focus on diverse patient populations, real-world environment testing, sensor improvement, and long-term rehabilitation evaluation. By addressing these issues, future research can provide more comprehensive, reliable, and applicable tools for physiotherapy movement assessment, benefiting clinical practice and patient rehabilitation. Therefore, future research should focus on the following directions:**Expanding Sample Diversity:** Future studies should include a more diverse range of participants, particularly patients with spinal cord injuries and varying cognitive impairments, to improve the generalizability and application value of the results.**Real-World Environment Validation:** Validation of the performance of algorithms and systems in real-world environments, such as homes and community settings, to ensure applicability.**Improving Sensor Technology:** Enhancement of the resolution and field of view of depth sensors, design of appropriate sensor application schemes, optimize motion capture performance, and development new sensors and improved algorithms to reduce environmental and wearable interference.**Developing Generalizable Algorithms:** Creation of algorithms that generalize well to dynamic movements, using large-scale, diverse datasets to improve model generalization.**Enhancing User Experience:** Improvement of virtual reality systems, the use of non-contact methods where possible to reduce patient discomfort, an increase in acceptability, and the design of systems and protocols that promote long-term home rehabilitation adherence.

### 4.4. Recommendation and Future Directions

With the continuous progress of computer vision technology, its application in camera depth sensor-based physiotherapy movement assessment also shows many promising research and development directions, although there are many shortcomings and limitations. Based on the previous analysis, future research should focus on the following aspects:

First, in-depth research is needed to investigate how to effectively apply depth camera technology in physical therapy rehabilitation exercises, especially with regards to data acquisition methods that do not require professional setup and are contactless,. In this way, it is possible to improve the algorithms to capture and analyze movement data in complex dynamic environments more accurately.

Secondly, developing lightweight and efficient algorithms for resource-constrained environments is also an important direction for future research. In terms of interdisciplinary collaboration, it is recommended that collaboration in computer science, physical therapy, neuroscience, and biomedical engineering be strengthened. Perspectives and methodologies from different fields can provide more comprehensive solutions that can drive innovation and application of physical therapy technologies.

In addition, existing machine learning and deep learning models need to be validated on more extensive and diverse datasets to improve their generalizability and robustness and explore new theoretical frameworks and algorithms to explain and optimize complex phenomena in movement capture and rehabilitation training, such as the limited application of self-supervised learning algorithms to the learning extraction of physical therapy movement feature data and the design of personalized physical therapy rehabilitation programs through downstream tasks.

Future research should explore the application of deep sensors in home rehabilitation, telemedicine, and community health centers, which will significantly enhance the accessibility and continuity of rehabilitation services, especially the development of rehabilitation devices and programs for home environments that can provide personalized and real-time feedback to improve patient engagement and rehabilitation outcomes. With our increasingly aging population, it is especially relevant to develop physical therapy smart applications for exercise and assessment for the elderly.

In summary, future research efforts should build on existing research by focusing on the full-scale application of deep cameras, the construction of adequate datasets, the application of self-supervised learning algorithms in physical therapy sports, as well as the exploration of new methods, the promotion of interdisciplinary collaborations, and the validation of theoretical hypotheses, with an eye on practical applications and emerging trends. By addressing these future directions, depth camera-based computer vision for physiotherapy movement assessment will continue to evolve, ultimately leading to more effective, convenient, and personalized patient care.

## 5. Conclusions

This systematic review examined machine learning-based computer vision techniques for depth camera-based physiotherapy movement assessment through analysis of 18 high-quality papers. The findings demonstrated both the potential and limitations of current approaches in this field.

The review revealed that depth cameras, particularly the Kinect family of sensors, dominated physiotherapy applications due to their ability to provide markerless, non-contact motion capture. The review identified three main data types used in the studies: RGB-D data (55.6%), skeletal data (27.8%), and multimodal data combining RGB-D with IMU measurements. Most studies employed the Kinect SDK 2.0 or OpenPose library for skeletal data extraction, supplemented by data alignment, normalization, and transformation techniques to enhance data utility.

From an algorithmic perspective, the field showed a balanced distribution between traditional machine learning methods (44.4%) and deep learning models (41.7%), with specialized dedicated algorithms accounting for the remaining 13.9%. Convolutional neural networks emerged as the predominant deep learning architecture, while random forest and support vector machines were the most commonly used traditional approaches.

The applications primarily focused on local environments (50%) followed by clinical settings (33.4%), with remote rehabilitation representing a smaller portion (22.3%). The research concentrated on assessing movements of the upper limbs, lower back, knees, and whole-body motion, with implementations varying across rehabilitation, assessment, and movement analysis applications.

Despite the promise of the technology, the translation of current methods into clinical practice faces multiple challenges:**Limitations of the validation population:** Existing studies are mainly based on small datasets with high participant homogeneity. Expanding the dataset to include participants of different ages, genders, and health statuses is critical to improving the model’s generalizability, genders, and health statuses.**Adaptability to the clinical environment:** Studies have been limited to controlled laboratory environments, making it challenging to fully reflect the complexities of clinical practice. Future research should focus on real-world validation to comprehensively assess the stability and reliability of the system in different application scenarios.**Complex dynamic motion analysis:** Existing algorithms still have limitations in handling dynamic and complex motion patterns. Integrating advanced technologies such as biomechanical modeling and real-time motion tracking is expected to break through this bottleneck.**Biomechanical holistic analysis:** An in-depth study of the biomechanical synergy between various body parts and movement’s coupling effect will help build a more comprehensive and accurate movement assessment model.**Insufficient data resources:** Current research relies on small and medium-sized datasets, which limits the performance and robustness of deep learning models. Establishing large-scale, multimodal, and well-labeled real-scene datasets will lay a more solid foundation for algorithm development and validation.

In order to bridge the gap from laboratory research to clinical practice, breakthroughs in sensor technology, algorithm development, and validation methods are needed. Future research should focus on developing cost-effective, easy-to-use systems that can be seamlessly integrated into clinical workflows. Interdisciplinary collaboration among physical therapists, engineers, and data scientists is essential to ensure the system’s utility, reliability, and fit with clinical needs.

Machine learning-based computer vision technology has the potential to fundamentally change the process of motion assessment in physical therapy by providing objective, efficient, and personalized assessment protocols. This study shows that although this technology shows great promise, sustained efforts are needed to apply it in clinical practice fully. The potential of such systems to improve rehabilitation outcomes and patient quality of life can only be fully realized if substantial progress is made in key technical areas.

## Figures and Tables

**Figure 1 sensors-25-01586-f001:**
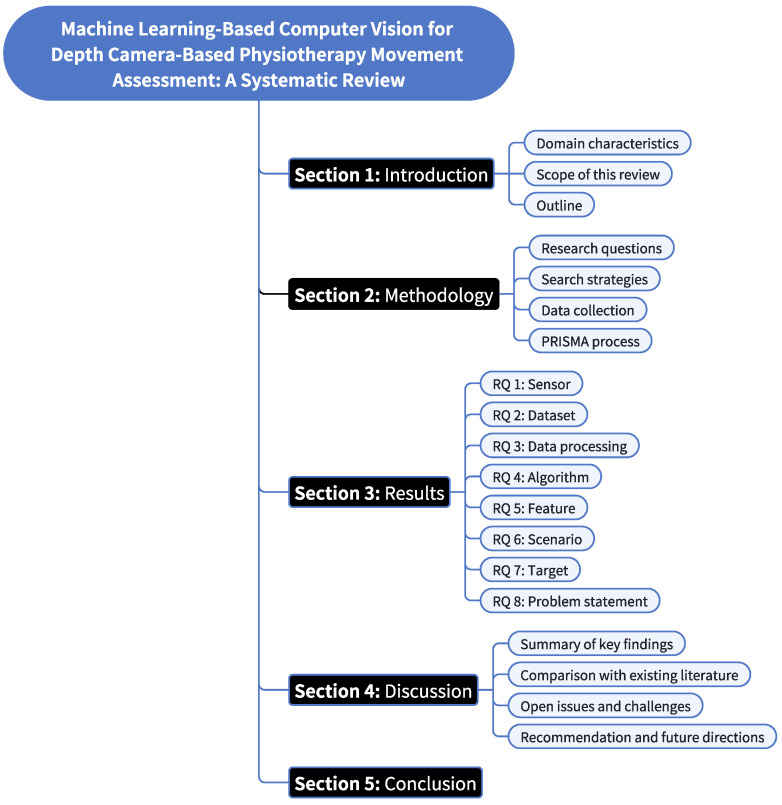
The structure of this article.

**Figure 2 sensors-25-01586-f002:**
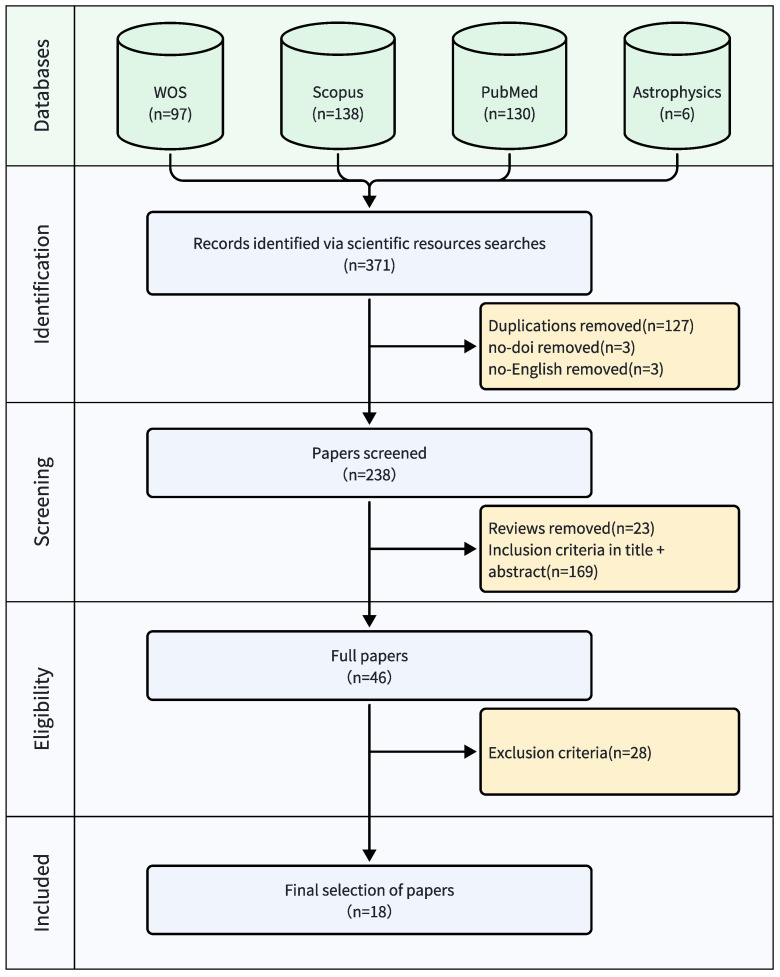
PRISMA diagram of the systematic review.

**Figure 3 sensors-25-01586-f003:**
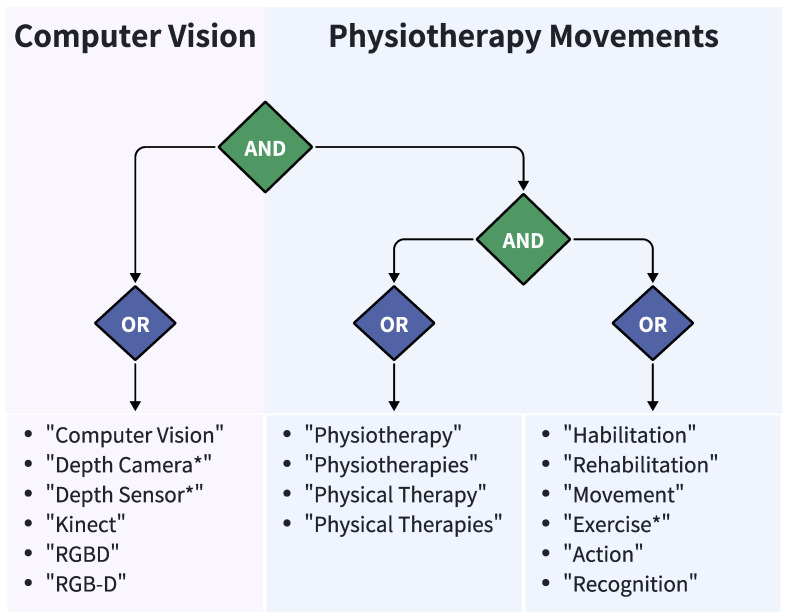
Boolean Search Strategy for Computer Vision and Physiotherapy Movements.

**Figure 4 sensors-25-01586-f004:**
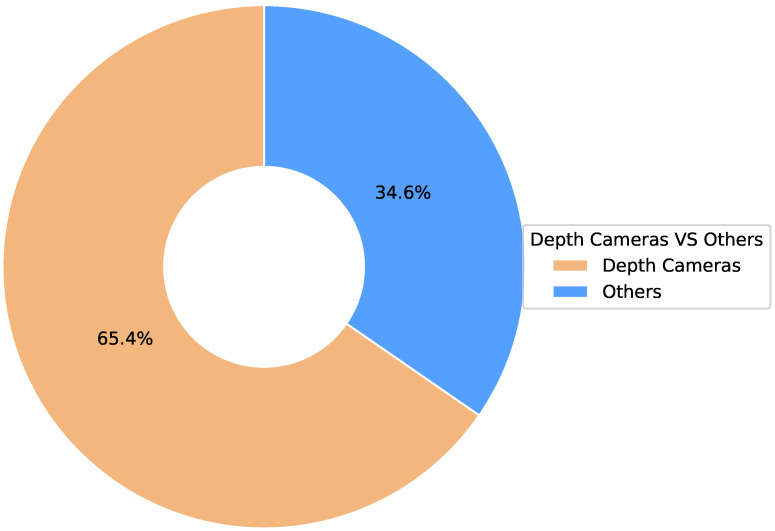
Sensor selection: depth-camera vs. other sensors.

**Figure 5 sensors-25-01586-f005:**
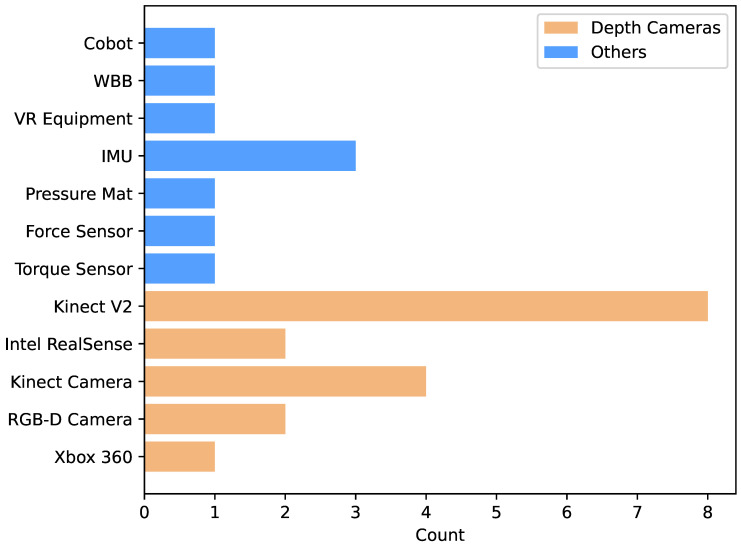
Survey of sensor in physiotherapy movement assessment.

**Figure 6 sensors-25-01586-f006:**
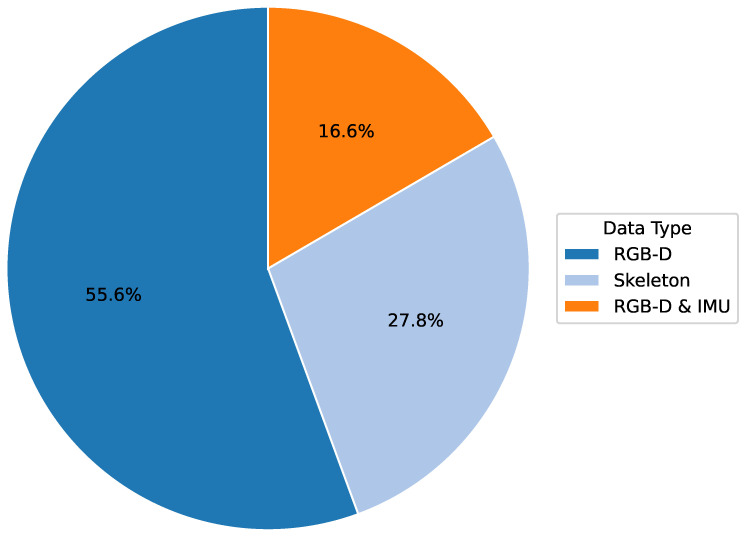
Pie chart of percentage distribution of data types.

**Figure 7 sensors-25-01586-f007:**
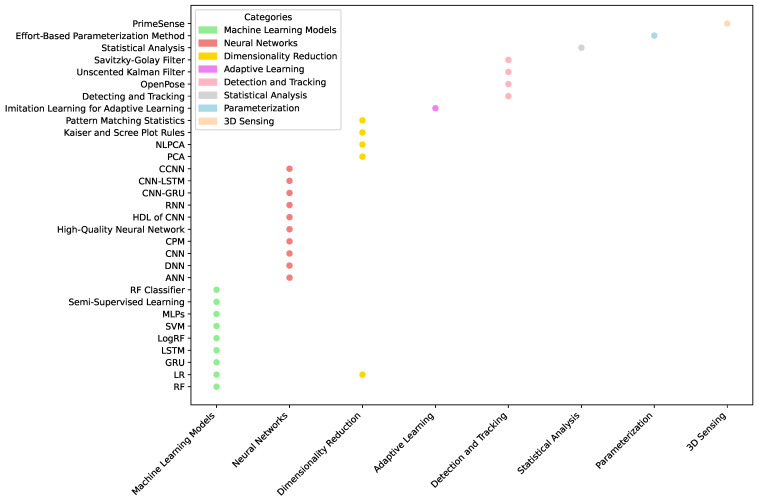
Applications and statistics of various algorithms in physiotherapy movement assessment.

**Figure 8 sensors-25-01586-f008:**
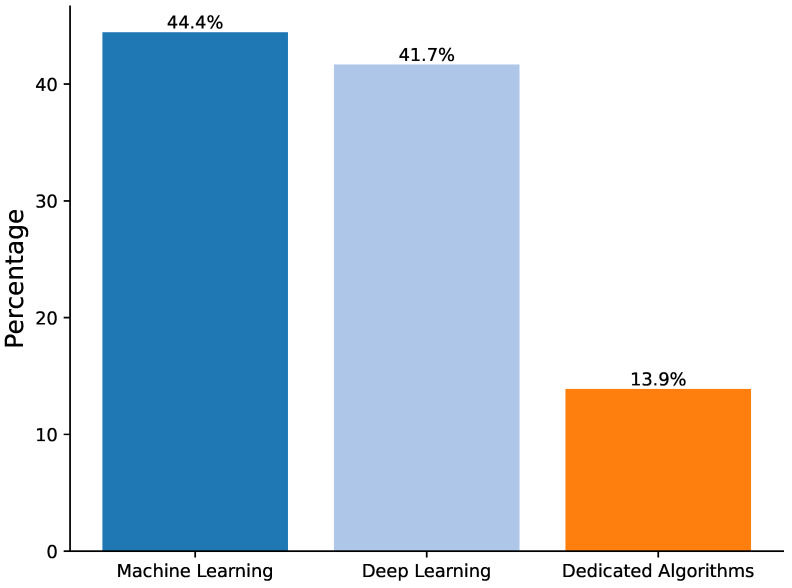
Diagram of bars for the algorithm in current literature.

**Figure 9 sensors-25-01586-f009:**
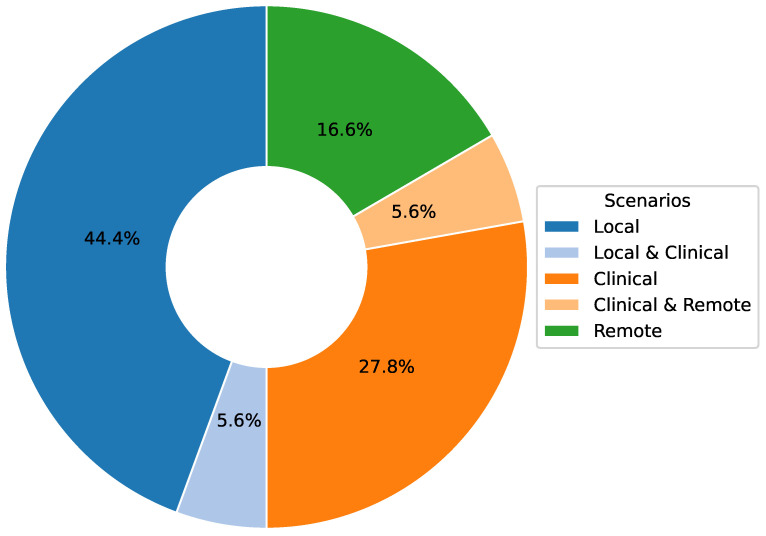
Pie chart depicting research application scenarios for physiotherapy movement assessment.

**Figure 10 sensors-25-01586-f010:**
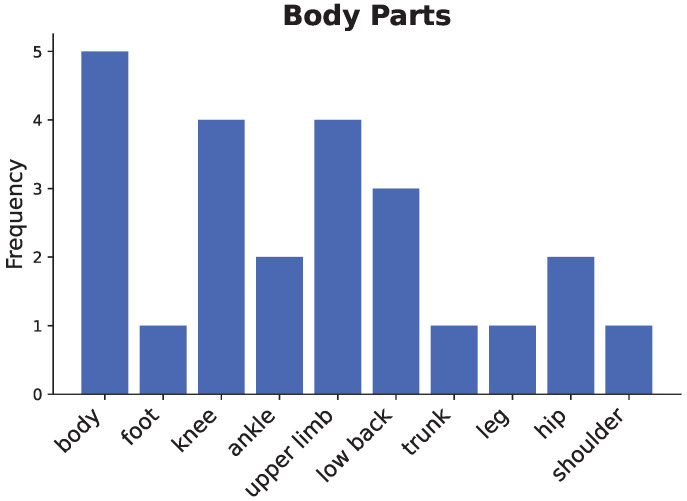
Diagram of bars for the studied body parts.

**Table 1 sensors-25-01586-t001:** Research questions and motivations for computer vision in camera-depth sensor-based physiotherapy movement assessments.

**RQ 1: What types of depth camera sensors are used in physiotherapy movement assessment, and what are their characteristics and applications?**
Motivation 1: Depth camera technologies (e.g., time of flight, structured light, and stereo vision) are essential to physiotherapy movement assessment. Each depth camera type has unique advantages, and choosing the right camera for a particular treatment scenario is critical. A comprehensive analysis of the depth cameras used in this study and their actual efficacy in physical therapy can provide an essential reference for practical applications to improve the evaluation system’s stability and performance, optimize the data acquisition quality, and enhance evaluation accuracy.
**RQ 2: How do the dataset type, construction methods, and feature selection impact algorithm performance and the effectiveness of applications in depth camera-based physiotherapy exercise assessments?**
Motivation 2: The quality and applicability of datasets are critical factors in determining the effectiveness of computer vision techniques in physiotherapy movement assessment. To systematically analyze the effects of dataset types, construction methods, and dataset sizes and the effectiveness of their application as used in the current literature to evaluate existing models’ reliability and provide methodological guidance for future research to optimize dataset construction strategies.
**RQ 3: What are the primary data processing methods used in depth-camera-based movement analysis for physical therapy, and how do they affect the assessment?**
Motivation 3: Data processing directly affects the assessment’s quality and accuracy. Systematic study of current data processing techniques (e.g., skeletal data extraction, coordinate system transformation, data normalization) can help optimize the processing flow, improve data quality, and enhance model generalization to improve the accuracy and clinical relevance of the assessment and drive depth camera-based movement analysis in physical therapy toward higher accuracy and a more comprehensive range of applications.
**RQ 4: Which algorithms perform best for depth camera-based physiotherapy movement assessment, and what are the advantages and limitations of different algorithms for various physical therapy tasks?**
Motivation 4: Algorithm selection is crucial for depth camera-based physiotherapy assessment, impacting accuracy, efficiency, and clinical utility. Comparing traditional machine learning, deep learning, and specialized algorithms across various physiotherapy tasks, elucidating their strengths and limitations. Optimizing algorithm choice aims to enhance assessment techniques, improve patient outcomes, and establish a robust foundation for intelligent, personalized physiotherapy interventions.
**RQ 5: What are the main innovative features of current research in physical therapy exercise assessment with depth camera sensors? How do these features improve rehabilitation outcomes and accessibility?**
Motivation 5: Depth camera sensor technology presents multiple innovative features in physiotherapy movement assessment, and systematic analysis of these features is essential to capture advances in the field and guide future research. By exploring how these innovations can work together to promote advances in physical therapy, comprehensive and effective rehabilitation systems could be developed, improving diagnostic accuracy and personalized rehabilitation outcomes while optimizing home rehabilitation and remote monitoring solutions.
**RQ 6: What are the main application scenarios for depth cameras in physical therapy, and how do these scenarios affect the implementation strategies and rehabilitation outcomes?**
Motivation 6: Depth camera technology is used in different scenarios, such as remote, clinical, and home settings. An in-depth understanding of the specific needs and challenges of these scenarios can help optimize the implementation strategy, improve the efficiency of rehabilitation resource allocation, and promote the development of physical therapy in the direction of more intelligent, personalized, and universal, thus improving the overall rehabilitation effect.
**RQ 7: What are the most common body parts targeted by camera depth sensor-based physiotherapy, and where is this technology most frequently implemented?**
Motivation 7: In physical rehabilitation, movement assessment on different body parts is critical to patient recovery. Clarifying the application of depth camera technology in assessing various body parts can help develop more comprehensive and accurate assessment methods, improve the relevance and effectiveness of rehabilitation treatment, and provide a reliable basis for developing individualized rehabilitation plans.
**RQ 8: What are the main challenges and limitations of computer vision and depth sensor technology in physiotherapy movement assessment?**
Motivation 8: Although depth camera technology has shown great potential in physical therapy, it still faces many challenges. A thorough analysis of these challenges will not only help to understand the limitations of the current technology but also point to future research that will lead to more accurate and reliable physiotherapy movement assessment systems and improved rehabilitation outcomes.

**Table 2 sensors-25-01586-t002:** Overview of Electronic Search Strategy: Including Databases, Search Queries.

Database	Query
WOS, PubMed	(”Computer Vision“ OR ”Depth Camera* “ OR ”Depth Sensor*“ OR ”Kinect“ OR ”RGBD“ OR ”RGB-D“) AND (”Physiotherapy“ OR ”Physiotherapies“ OR ”Physical Therapy” OR “Physical Therapies”) AND (“Habilitation” OR “Rehabilitation” OR “Movement” OR “Exercise*” OR “Action” OR “Recognition”)
Scopus	TITLE-ABS-KEY ((“Computer Vision” OR “Depth Camera*” OR “Depth Sensor*” OR “Kinect” OR “RGBD” OR “RGB-D”) AND (“Physiotherapy” OR “Physiotherapies” OR “Physical Therapy” OR “Physical Therapies”) AND (“Habilitation” OR “Rehabilitation” OR “Movement” OR “Exercise*” OR “Action” OR “Recognition”))
Astrophysics	(title:“Computer Vision” OR title:“Depth Camera*” OR title:“Depth Sensor*” OR title:“Kinect” OR title:“RGBD” OR title:“RGB-D” OR keyword:“Computer Vision” OR keyword:“Depth Camera*” OR keyword:“Depth Sensor*” OR keyword:“Kinect” OR keyword:“RGBD” OR keyword:“RGB-D” OR abstract:“Computer Vision” OR abstract:“Depth Camera*” OR abstract:“Depth Sensor*” OR abstract:“Kinect” OR abstract:“RGBD” OR abstract:“RGB-D”) AND (title:“Physiotherapy” OR title:“Physiotherapies” OR title:“Physical Therapy” OR title:“Physical Therapies” OR keyword:“Physiotherapy” OR keyword:“Physiotherapies” OR keyword:“Physical Therapy” OR keyword:“Physical Therapies” OR abstract:“Physiotherapy” OR abstract:“Physiotherapies” OR abstract:“Physical Therapy” OR abstract:“Physical Therapies”) AND (title:“Habilitation” OR title:“Rehabilitation” OR title:“Movement” OR title:“Exercise*” OR title:“Action” OR title:“Recognition” OR keyword:“Habilitation” OR keyword:“Rehabilitation” OR keyword:“Movement” OR keyword:“Exercise*” OR keyword:“Action” OR keyword:“Recognition” OR abstract:“Habilitation” OR abstract:“Rehabilitation” OR abstract:“Movement” OR abstract:“Exercise*” OR abstract:“Action” OR abstract:“Recognition”) AND (pubdate: [2020-01-01 TO 2024-12-31])

**Table 3 sensors-25-01586-t003:** The final selection of articles for the review.

ID	Author	Year	Country	Database	Summary
1	Lim et al. [15]	2024	Canada	WOS, PubMed	Feasibility of depth cameras & pressure pads as alternatives to force plates.
2	Wagner et al. [16]	2023	Poland	PubMed	Depth-sensor gait methods compared.
3	Raza et al. [27]	2023	Pakistan, Saudi Arabia	WOS, Scopus	AI for pose estimation in physiotherapy exercises.
4	Maskeliünas et al. [10]	2023	Lithuania	WOS, Scopus	BiomacVR for posture & movement analysis in rehabilitation.
5	Lim et al. [11]	2023	China	Scopus, PubMed	Adaptive Cobot system for assistive rehab training.
6	Khan et al. [28]	2023	USA	WOS	Quantum neural network for post-stroke exercise assessment.
7	Bijalwan et al. [29]	2023	India	WOS, Scopus	Automated system for upper limb exercise detection using an RGB-Depth camera.
8	Keller et al. [30]	2022	USA	PubMed	Unsupervised ML for low back pain exercise strategies.
9	Zhao et al. [31]	2021	USA, China	WOS, Scopus	Home TKR rehab system development.
10	Trinidad-Fernandez et al. [32]	2021	Spain, Belgium	PubMed	RGB-D camera validates motion capture in spondyloarthritis.
11	Hustinawaty et al. [33]	2021	Indonesia	Scopus	Kinect SDK for a study of straight leg lift exercise.
12	Girase et al. [34]	2021	USA	PubMed	Key factors identified for spine, hip, and knee assessment from sit-to-stand.
13	Çubukçu et al. [12]	2021	Turkey	WOS, Scopus	Kinect-based mentor for shoulder injury telerehab.
14	Wei et al. [35]	2020	USA	WOS, Scopus	Sensors and DL for automated balance assessment.
15	Uccheddu et al. [36]	2021	Italy	WOS, Scopus	Hybrid approach for 3D pose estimation proposed.
16	Trinidad-Fernandez et al. [37]	2020	Belgium, Spain, Australia	PubMed	RGB-D camera kinematic assessment results.
17	Saratean et al. [13]	2020	Romania	WOS, Scopus	Kinect-based physical therapy guidance system.
18	Garcia et al. [8]	2020	Brazil	WOS, Scopus	RGB-D camera analysis of compensatory trunk movements.

**Table 4 sensors-25-01586-t004:** Summary of Target and Sensor in Physiotherapy Movement Assessment.

**ID**	**Target**	**Sensor**
1	body	Kinect V2, Pressure Mat
2	foot, knee, ankle	Kinect V2
3	body	Ordinary Camera
4	upper limb	Intel RealSense L515/D435i, HTC Vive VR Equipment
5	upper limb	Kinect Camera, Cobot, Force/Torque Sensor
6	body	Kinect
7	upper limb	Kinect V2
8	low-back	Kinect V2
9	knee	Kinect V2, IMU(Shimmer)
10	trunk	RGB-D Camera, IMU(MP67B)
11	leg	Kinect
12	low-back, hip, knee	Kinect V2
13	shoulder	Kinect V2
14	body	Kinect, WBB
15	hip, knee, ankle	Intel RealSense D415
16	low-back	RGB-D Camera (Xtion Pro), IMU(MP67B)
17	body	Kinect for Xbox 360
18	upper limb	Kinect V2

**Table 5 sensors-25-01586-t005:** Summary of Data Types and Datasets in Physiotherapy Movement Assessment.

**ID**	**Data Type**	**Dataset**
1	Joint displacement data series	10 non-disabled participants: 7 males, 3 females
2	RGB-D Images	5 subjects: 2 males, 3 females
3	Skeleton Data	Multi-Class Exercise Poses for Human Skeleton
4	RGB-D Videos	16 healthy subjects, 10 post-stroke patients
5	RGB-D Image	5 healthy subjects
6	Joint-Skeletal	UI-PRMD
7	RGB-D	UTD-MHAD, mHealth, OU-ISIR, HAPT
8	RGB-D	111 participants: back pain 43, control 26, surgery 4
9	RGB-D & IMU	/
10	RGB-D Videos & IMU	17 subjects: 54.35 (±11.75) years
11	RGB-D	10 human objects
12	RGB-D Time Series	3 patient groups and one control group: 78 control, 130 LBP, 90 hip, and 113 knee
13	Skeleton Data	29 shoulder damaged volunteers: 18 males, 11 females
14	RGB-D Image	41 subjects: 26 males, 15 females; 21 healthy subjects and 20 patients with PD
15	RGB-D Videos	/
16	RGB-D & IMU	30 subjects: 18 65 years with non-specific lumbar pain
17	Skeleton Data	/
18	RGB-D	14 volunteers: 9 range of movement capture tests, 5 trunk compensation tests

**Table 6 sensors-25-01586-t006:** Summary of Algorithm and Processing in Physiotherapy Movement Assessment.

ID	Algorithm	Processing
1	/	Joint Displacement Data.
2	Savitzky-Golay Filter	Transform the coordinate system using the KD, CH, and FV data processing methods.
3	RF, LR, GRU, LSTM, LogRF	MediaPipe Pose Marker, Feature Selection, and Hyperparameter Tuning.
4	ANN, DNN, CNN, CPM	Human skeletal movement was observed using visible information.
5	Imitation Learning for Adaptive Learning	/
6	High-Quality Neural Network	Align the length and center and perform characteristic transformation.
7	HDL of CNN, RNN, CNN-GRU, and CNN-LSTM	Apply Min-Max normalization.
8	PCA, NLPCA, LR, Kaiser and Scree Plot Rules, Pattern Matching Statistics	Use Kalman filter, sequential second-order, and low pass Butterworth filtering.
9	/	Fuse accelerometer and gyroscope measurements.
10	Statistical analysis	Synchronize the dataset with the timestamp and visualize it using OpenNI2, NiTE2, and MRPT.
11	Detecting and Tracking	Calibration, skeletalization process, and feature extraction.
12	SVM, RF, MLPs, CCNN, Semi-Supervised Learning, Unscented Kalman Filter	Estimate joint center positions using the standard Kinect 2 Body Tracking library.
13	Statistical Analysis	Use the Kinect SDK 2.0.
14	CNN, RF Classifier	Perform data augmentation.
15	OpenPose	Process video frames from the RGB sensor with the OpenPose library.
16	/	Synchronize and use OpenNI2 and NiTE2 to create a virtual skeleton representation.
17	Effort-Based Parameterization Method	/
18	PrimeSense	Use the Kinect SDK 2.0.

**Table 7 sensors-25-01586-t007:** Comparison of Algorithm Categories.

Algorithm Type	Performance	Accuracy	Limitations	Interpretability
Traditional ML	High for small datasets	47.2–99.8%	Limited for high-dimensional, nonlinear data	High due to clear feature contributions
Deep Learning	Computationally intensive	60.7–100%	Requires large labeled datasets, ”black-box“ nature	Low, needs explainable AI (XAI) techniques
Dedicated	Optimized for niche tasks	51.7 – 81.8%	Limited generalizability	High, tailored to specific applications

**Table 8 sensors-25-01586-t008:** Summary of the Research Features.

ID	Feature
1	Demonstrates integration of depth cameras and pressure mats as cost-effective, accessible feedback mechanisms for balance training.
2	Advances gait assessment techniques using depth sensor data, improving diagnostic and treatment accuracy.
3	Uses the LogRF method and random forest algorithms for improved human pose estimation in physiotherapy.
4	Employs virtual reality to increase precision and engagement in posture and movement analysis during rehabilitation.
5	Features a personalized adaptive learning system for upper-limb rehabilitation, enhancing patient-specific outcomes.
6	Introduces a hybrid quantum neural network to enhance speed and accuracy in post-stroke exercise assessments.
7	Combines deep learning models to enhance modeling of spatio-temporal features in stroke rehabilitation.
8	Utilizes unsupervised machine learning to analyze movement capture data, identifying movement strategies in low back pain patients.
9	Proposes a protocol for home-based rehabilitation post-knee replacement using accessible technology.
10	Uses an RGB-D camera to accurately assess movement limitations in spondyloarthritis, supporting better clinical decisions.
11	Applies Kinect SDK skeletonization to accurately assess the straight leg raise, aiding lumbar condition diagnoses.
12	Automates detection and classification of pathologies from sit-to-stand movements using machine learning.
13	Develop a system using Kinect to monitor and correct shoulder exercises dynamically.
14	Integrates sensors and deep learning to enable real-time balance evaluations, enhancing therapy effectiveness.
15	Develop a hybrid method using RGB-D sensors for accurate joint angle estimation in-home rehabilitation.
16	Validates the use of RGB-D cameras for reliable and responsive kinematic assessments in clinical settings.
17	Implements a Kinect-based system for remote physiotherapy coaching, facilitating continuous care.
18	Analyze compensatory trunk movements with RGB-D cameras to refine upper limb rehabilitation strategies.

**Table 9 sensors-25-01586-t009:** Summary of Scenarios in Physiotherapy Movement Assessment.

**ID**	**Scenario**	**Objective**
1	Local	Evaluate balance training effectiveness with depth cameras and pressure mats.
2	Local	Enhance gait analysis accuracy using new spatiotemporal methods and depth sensors.
3	Local	Improve exercise correction in physiotherapy with innovative pose estimation.
4	Remote, Clinical	Develop a VR system for precise human posture and motion analysis to boost rehabilitation engagement.
5	Remote	Build a personalized adaptive learning system with collaborative robots for upper-limb rehab.
6	Local	Improve post-stroke exercise assessments with a hybrid quantum neural network.
7	Clinical	Enhance upper extremity rehab post-stroke by modeling spatio-temporal features with deep learning.
8	Clinical	Discover low back pain strategies using unsupervised learning on motion data.
9	Local	Establish a comprehensive home protocol for post-knee replacement recovery.
10	Clinical	Validate RGB-D cameras for precise trunk movement analysis in spondyloarthritis.
11	Clinical, Local	Analyze straight leg raises accurately using Kinect SDK’s skeletonization.
12	Clinical	Automate diagnosis of spinal, hip, and knee pathologies from sit-to-stand movements.
13	Remote	Develop a Kinect-based system to monitor and correct shoulder rehab exercises.
14	Local	Develop an on-demand balance evaluation tool integrating sensors with deep learning.
15	Local	Merge 2D and 3D RGB-D data for precise joint angle estimation in-home rehab.
16	Local	Validate the reliability and responsiveness of kinematic assessments with RGB-D cameras.
17	Remote	Implement a Kinect-based remote physiotherapy coaching system to ensure exercise adherence.
18	Clinical	Analyze compensatory trunk movements in upper limb rehab using RGB-D cameras.

**Table 10 sensors-25-01586-t010:** Summary of the Research Problem Statement.

**ID**	**Problem Statement**
1	Current feedback mechanisms in balance training are restricted by their reliance on expensive, bulky equipment.
2	Traditional gait analysis needs to harness depth sensor data, impacting diagnostic accuracy effectively.
3	Current pose estimation in physiotherapy often lacks precision, leading to ineffective exercise correction.
4	Conventional motion analysis tools lack the precision and interactivity required for effective rehabilitation.
5	Standard upper-limb rehabilitation devices do not adapt to patient progress, limiting their effectiveness.
6	Existing post-stroke assessments lack precision and speed, necessitating advanced computational solutions.
7	Spatio-temporal feature modeling in stroke rehabilitation is inadequate, hindering exercise effectiveness.
8	Personalized treatment strategies in low back pain are limited by poor analysis of movement data.
9	Spatio-temporal feature modeling in stroke rehabilitation is inadequate, hindering exercise effectiveness.
10	Tools for assessing movement limitations in spondyloarthritis need to be improved.
11	Current methods need to capture the straight leg raise, complicating lumbar assessments accurately.
12	Automated diagnostic tools for spine, hip, and knee pathologies need to be improved.
13	Existing shoulder rehab systems lack precise, interactive monitoring of exercises.
14	Current physical therapy lacks practical on-demand balance evaluation tools.
15	Home rehab methods inaccurately estimate joint angular ranges, affecting treatment outcomes.
16	Kinematic assessments lack the reliability and responsiveness required for effective clinical decisions.
17	Current physical therapy lacks effective on-demand balance evaluation tools.
18	Methods to analyze compensatory trunk movements in upper limb rehab are ineffective.

**Table 11 sensors-25-01586-t011:** Summary of the Research Issues and Challenges.

**ID**	**Ref**	**Limitation**
1	[15]	The study only tested non-disabled participants, not spinal cord injury patients.
2	[16]	Gait analyses were conducted under controlled lab conditions, which may not reflect real-world variability.
3	[27]	Pose estimation algorithms tested primarily in well-controlled environments, may not perform as well in cluttered spaces.
4	[10]	VR systems may cause discomfort or dizziness in some patients, limiting their widespread usability.
5	[11]	System’s adaptability is not tested on patients with varying degrees of cognitive impairments.
6	[28]	Validation is restricted to small datasets which may not generalize to broader populations.
7	[29]	Deep learning models require extensive computational resources, limiting deployment in low-resource settings.
8	[30]	Machine learning models derived from a limited demographic, potentially affecting the universality of findings.
9	[31]	Study did not account for long-term adherence to home-based programs.
10	[32]	Camera’s depth resolution is insufficient to capture fine-grained joint movements accurately.
11	[33]	Kinect SDK’s accuracy in capturing leg movements is not verified against gold-standard clinical assessment tools.
12	[34]	Diagnostic accuracy is dependent on the precise execution of sit-to-stand movements, which varies widely among patients.
13	[12]	Limited to static exercises; dynamic movements’ complexity not fully explored.
14	[35]	Balance evaluation algorithms not validated in diverse real-world environments.
15	[36]	Estimations may not be accurate for patients with severe joint deformities or those wearing certain types of clothing that interfere with sensor accuracy.
16	[37]	Kinematic data’s reliability is compromised by occasional sensor inaccuracies and environmental interferences.
17	[13]	Kinect sensor’s limited field of view can restrict the range of exercises that can be monitored.
18	[8]	Analysis does not account for simultaneous lower limb movements, which can influence trunk motion.

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
