# Peer review of "Machine Learning-Based Computer Vision for Depth Camera-Based Physiotherapy Movement Assessment: A Systematic Review"

_sensors, 2025, doi:10.3390/s25051586_

Round 1

Reviewer 1 Report

Comments and Suggestions for Authors

This review paper systematically studied the literature on depth camera-based movement assessment, methods, datasets, ML architectures, and computer vision techniques in physiotherapy applications. With a few revisions to enhance clarity and focus, the manuscript can be a valuable resource for researchers in this field.

1.     The paper is generally well-written, well-organized, and easy to follow, but its structure still can be improved. For instance, in the section discussing the distribution of research focus (lines 673-675), the concluding sentence seems disconnected from the body of that paragraph.

2.     As a minor suggestion, some concluding parts of sections do not provide new insights. For example, lines 659-661 or 332-334 do not contribute significant information or insight from the reviewers to the authors.

3.     You may have missed some relevant keywords including “kinematics”, “pose estimation”, “posture”, and so on. So, you may have skipped some notable related literature during the initial query process. So, please double-check for relevant keywords that might have excluded notable studies.

4.     RQ4 algorithms Section (3.4): This is one of the most critical sections of the paper, but it would benefit from additional structure and detail. Consider summarizing results into a table, similar to the other sections, rather than the figure currently included. Also, you may provide insights into the performance metrics (accuracy, precision, recall) reported in these studies. When you mention accurate/precise in this section it is not clear that you are talking about how much accuracy. Minor: the concluding statement of this section (lines 406 - 410) is also general and you may consider rewriting it.

5.     You could share your critiques/insights/suggestions about using RealSense vs Kinect on lines 244-248. You may discuss scenarios where each sensor is more suitable and why, such as specific rehabilitation tasks, data requirements, or even cost considerations.

6. The quality of Figure 4 can be improved. Consider reproducing it for better readability and presentation.

Thank you.

Author Response

Thank you very much for taking the time to review this manuscript. We sincerely thank you for your valuable comments and suggestions on this paper. Your meticulous review and insightful suggestions have enabled us to think deeply and improve the content of the paper from multiple perspectives. We have carefully read each and every one of the comments and made comprehensive revisions and improvements to the paper with the attitude of striving for excellence. The main revisions are reported as follows:

  1. According to the experts' suggestions, we focused on optimizing the overall structure of the thesis, especially on the systematic rewriting of the chapters, such as the distribution of research focuses, and tried to enhance the logical coherence between paragraphs. At the same time, we have also improved the chapters on algorithms, datasets, sensors, and targets and tried to make the text more fluent and natural.

  1. In response to the depth of insights raised by the experts, we have made careful revisions to the summary parts of the chapters. In particular, we have added more in-depth research findings and their potential significance in the contents of pages 28-29 (lines 787-802) and page 16 (lines 367-373).

  1. Regarding the completeness of the search terms, we accepted the experts' suggestions and conducted additional tests in the Web of Science database. The test results show that the addition of the keywords "kinematics", "pose estimation" and "posture" have less impact on the literature search results, which, to a certain extent, verifies the reasonableness of the original keywords.

  1. For the improvement suggestions in the algorithm section (pp. 16-20, lines 375-519), we added new tables and charts and endeavored to systematically compare and analyze the algorithms in multiple dimensions, such as performance metrics, application scenarios, strengths and weaknesses, and ethical considerations, etc., in an attempt to provide readers with more comprehensive references.

  1. In the section on comparative analysis of sensors (page 12, lines 271-283), we have supplemented in detail the characteristics and applicability of different sensors, such as RealSense, Kinect, etc., in rehabilitation scenarios according to the experts' opinions, with a view to providing more valuable references for practical applications.

  1. Following the experts' suggestions, we redrew Figure 4, striving to improve the quality and readability of the pictures and present clearer visual effects to the readers.

Reviewer 2 Report

Comments and Suggestions for Authors

The manuscript under review presents a systematic exploration of the application of machine learning-based computer vision techniques in the context of depth camera-enabled physiotherapy movement assessment. While it exhibits several commendable aspects, there are also certain areas that warrant further refinement and elaboration.

1) It is recommended that the abstract be written in one paragraph.

2) The introduction establishes the pertinence of the research topic within the realm of physiotherapy and computer vision. However, to enhance its impact, including more tangible real-world case studies or pilot applications would provide a more vivid illustration of the practical implications and exigency of such research. This would not only engage the reader more effectively but also fortify the motivation for the study.

3) The research questions are formulated with a reasonable degree of comprehensiveness. Nevertheless, certain questions, such as RQ4, could benefit from a more granular definition. For instance, when inquiring about the optimal-performing algorithms, a more explicit specification of the evaluation metrics (e.g., precision, recall, F1-score in addition to accuracy) and the specific physiotherapy task constraints (e.g., real-time processing requirements, limited data availability scenarios) would render the question more amenable to a focused and incisive analysis.

4) The adherence to the PRISMA guidelines and the detailed description of the data collection process are laudable, enhancing the methodological rigor and reproducibility of the study. However, a more elaborate account of the data extraction procedures, including the utilization of standardized data extraction forms or software tools, would augment the transparency and reliability of the process. Additionally, a more in-depth discussion of any inter-rater reliability issues or discrepancies encountered during data extraction and the strategies employed to resolve them would be instructive.

5) The dataset analysis is comprehensive, yet the issue of dataset standardization and interoperability remains relatively unexplored. The lack of standardized datasets may impede the direct comparison and meta-analysis of research findings, hindering the consolidation of knowledge in the field. Moreover, a more in-depth exploration of the challenges associated with annotating complex physiotherapy movements, such as the inter-rater variability and the impact of annotation errors on algorithm performance, would be a welcome addition.

6) The algorithm overview is instructive, yet a more in-depth comparative analysis of algorithm performance, especially in the context of head-to-head evaluations on standardized datasets and under clinically relevant constraints, would be highly beneficial. Additionally, the issue of algorithm interpretability, a critical factor in the clinical acceptance and trustworthiness of these techniques, requires more in-depth consideration. Methodologies such as explainable AI techniques could be explored to shed light on the inner workings of these algorithms.

7) The body part targeting analysis is useful, but an exploration of the biomechanical synergies and kinematic couplings between different body parts during physiotherapy movements could provide a more holistic understanding. This could potentially inform the development of more integrated and comprehensive movement assessment models.

8) In conclusion, while the manuscript represents a significant contribution to the field, the aforementioned areas of improvement could enhance its overall quality and impact, thereby providing a more robust foundation for future research and clinical translation in the area of machine learning-based computer vision for physiotherapy movement assessment.

9) The problem statement is comprehensive, yet a more proactive approach in proposing potential research avenues and technological innovations to surmount these challenges would be highly instructive. For example, the exploration of emerging sensor technologies with enhanced resolution and miniaturization capabilities or the development of novel machine learning architectures tailored to the unique characteristics of physiotherapy data could be proposed.

Author Response

Thank you very much for taking the time to review this manuscript. We sincerely appreciate the experts’ valuable comments and suggestions, which helped us identify shortcomings and improve the paper. Below is a summary of the main revisions:

  1. Abstract (Page 1, Lines 1–14): Reorganized and integrated into a single paragraph to enhance coherence and readability while preserving the core information.
  2. Introduction (Pages 1–2, Lines 25–36, 69–82): Added practical examples, such as post-stroke rehabilitation (Maskeliunas et al.) and visual feedback balance training (Lim et al.), highlighting the practical value of computer vision in physical therapy.
  3. Methodology (Pages 4–5, Lines 140–144): Elaborated on data extraction processes, including standardized forms and team validation, to improve transparency and reproducibility.
  4. Dataset Analysis (Pages 13–14, Lines 304–328): Analyzed key challenges like data standardization, interoperability, and action annotation to provide valuable references for future research.
  5. Algorithms (Pages 16–20, Lines 375–519): Enhanced analysis with performance comparisons, scenario evaluations, and discussions on interpretability and generalization.
  6. Goal Analysis (Page 26, Lines 680–702): Expanded discussions on biomechanical synergies and motor coupling in physical therapy to strengthen the theoretical foundation.
  7. Conclusion (Pages 33–34, Lines 982–1024): Improved systematic analysis of findings and added practical recommendations for future research.
  8. Problem Statement (Page 28, Lines 744–765): Introduced forward-looking research directions, including differential privacy, federated learning, and cryptographic computing.

These improvements, guided by expert feedback, have significantly enhanced the paper’s academic quality. We remain open to further suggestions to achieve even higher standards. Thank you again for your invaluable support.

Round 2

Reviewer 2 Report

Comments and Suggestions for Authors

The revised paper is approved for publication.